# 3DS: Decomposed Difficulty Data Selection's Case Study on LLM Medical Domain Adaptation

## Abstract

Large Language Models (LLMs) excel in general tasks but struggle in specialized domains like healthcare due to limited domain-specific knowledge. Supervised Fine-Tuning (SFT) data construction for domain adaptation often relies on heuristic methods, such as GPT-4 annotation or manual data selection, with a data-centric focus on presumed diverse, high-quality datasets. However, these methods overlook the model's inherent knowledge distribution, introducing noise, redundancy, and irrelevant data, leading to a mismatch between the selected data and the model's learning task, resulting in suboptimal performance. To address this, we propose a two-stage *model-centric* data selection framework, **Decomposed Difficulty Data Selection (3DS)**, which aligns data with the model's knowledge distribution for optimized adaptation. In **Stage 1**, we apply Prompt-Driven Data Selection via Explicit Alignment, where the model filters irrelevant or redundant data based on its internal knowledge. In **Stage 2**, we perform Decomposed Difficulty Data Selection, where data selection is guided by our defined difficulty decomposition, using three metrics: *Instruction Understanding*, *Response Confidence*, and *Response Correctness*. Additionally, an *attention-based importance weighting mechanism* captures token importance for more accurate difficulty calibration. This two-stage approach ensures the selected data is not only aligned with the model's knowledge and preferences but also appropriately challenging for the model to learn, leading to more effective and targeted domain adaptation fine-tuning. In the case study of the medical domain, our extensive experiments on real-world healthcare datasets demonstrate the superiority of 3DS over existing methods in accuracy by over 5.29%. Our dataset and code will be open-sourced at `https://anonymous.4open.science/r/3DS-E67F`.

## 1 Introduction

**Large Language Models (LLMs)** like GPT-4 (OpenAI, 2023) have showcased significant potential in natural language understanding. Open-source models such as LLaMA (Touvron et al., 2023) and Qwen (Bai et al., 2023) have also rapidly advanced, delivering competitive performance. However, in specialized domains like healthcare, their effectiveness is often constrained by the lack of domain-specific knowledge (Sanaei et al., 2023; Harris, 2023; Waisberg et al., 2023), essential for tasks like diagnosis (Panagoulias et al., 2024; Ullah et al., 2024) and treatment recommendations (Wilhelm et al., 2023; Nwachukwu et al., 2024). To address this, some works (Wang et al., 2023a; Zhang et al., 2023; Yang et al., 2023b; Zhu et al., 2023a; Pal & Sankarasubbu, 2023) have adapted LLMs to the medical domain by training on large-scale healthcare-specific datasets.

A common approach for LLM domain adaptation is **Supervised Fine-Tuning (SFT)** on domain instruction tuning datasets. Unlike continued pre-training, where data quantity is crucial (Que et al., 2024), existing works (Zhou et al., 2024) show that SFT requires only a *small but high-quality* dataset to effectively trigger a model's abilities in the desired direction. Expanding the dataset without careful selection can introduce challenges that affect model performance (Wang et al., 2023d), highlighting the need for additional factors in data selection to ensure effective fine-tuning. Yet, it remains unclear how to define optimal data samples for instruction tuning and systematically identify them. Efforts have largely relied on heuristic methods, such as GPT-4 annotation (Liu et al.,

2023) or manual data selection (Ji et al., 2023; Song et al., 2024), taking a data-centric approach that prioritizes what is assumed to be diverse and high-quality datasets. However, these datasets may fail to align with the model's actual needs, **creating gaps between the selected data and the model's inherent knowledge**, where the inherent knowledge refers to the broad, task-agnostic factual knowledge embedded in the model's parameters during pre-training on large, diverse textual corpora (Petroni et al., 2019; Cohen et al., 2023; AlKhamissi et al., 2022). Training on data that fails to align with this distribution can lead to suboptimal fine-tuning performance (Gekhman et al., 2024; Ren et al., 2024). To bridge this gap, we hope to explore a model-centric approach that focuses on effectively selecting data aligned with the model's current knowledge distribution. We define this setting as *model-centric instruction data selection*:

*Given a general LLM and a large domain-specific instruction dataset, how can we efficiently select data based on the model's knowledge distribution to best trigger its domain abilities?*

We address this problem by aligning training data with the model's inherent knowledge distribution, optimizing both its **informativeness and complexity** to drive effective learning. This alignment ensures the model is exposed to data that is both engaging and appropriately challenging, allowing it to build on existing knowledge while addressing gaps. As a result, two key challenges emerge:

***C1. How to filter low-quality and redundant data for efficient and effective domain adaptation?***
Domain-specific datasets, aggregated from diverse, large-scale sources, often contain noisy or redundant data—reintroducing knowledge the model has already internalized. Such data can disrupt learning (Wang et al., 2024a), hinder the identification of knowledge gaps (Havrilla & Iyer, 2024), waste resources, and increase the risk of overfitting (Budach et al., 2022; Wang et al., 2024b). In domain adaptation, acquiring specialized knowledge makes a model-centric data selection strategy necessary. The strategy must filter data based on the model's internal knowledge, dynamically removing redundancy and noise to focus on novel, challenging tasks.

***C2. How to balance data difficulty with the model's learning capacity?*** Data difficulty refers to the degree to which the model has mastered the data. The difficulty of domain-specific instruction data plays a critical role in shaping the model's learning. Overly simple data wastes resources, while overly complex data can overwhelm the model and stall progress (Gekhman et al., 2024; Ren et al., 2024; Kang et al., 2024). In domain adaptation, dynamically adjusting data selection to match the model's evolving knowledge is both essential and challenging. Accurately assessing the model's current state and determining how it handles increasing data complexity is difficult. The variability in learning progress further complicates efforts to calibrate data difficulty, making it challenging to avoid under- or overloading the model. Achieving this balance is key to ensuring steady learning and maximizing domain-specific knowledge acquisition.

To address these challenges, we propose **D**ecomposed **D**ifficulty **D**ata **S**election (3**DS**), a two-stage model-centric data selection framework which aligns with the model's knowledge distribution to optimize domain adaptation. **1)** For **C1**, we employ *Prompt-Driven Data Selection via Explicit Alignment*, where the model scores the dataset to explicitly remove irrelevant information. This ensures that only high-quality data aligned with the model's internal knowledge and preferences (model's judgment to decide what data is good) is retained, minimizing noise. **2)** To address **C2**, we propose a novel *Decomposed Difficulty Data Selection via Implicit Distribution Modeling*. This approach extends traditional perplexity (PPL) calculations by introducing three key difficulty metrics: ***Instruction Understanding Difficulty***, ***Response Confidence Difficulty***, and ***Response Correctness Difficulty***. Additionally, we apply an ***attention-based importance weighting mechanism*** to capture the varying importance of tokens, ensuring more accurate difficulty evaluation. This method ensures that data complexity is dynamically aligned with the model's learning capacity, optimizing the fine-tuning process. In summary, our contributions are as follows:

- We introduce 3DS, a two-stage model-centric data selection framework aligning training data with the model's inherent knowledge distribution, optimizing effective domain adaptation.
- We propose a novel difficulty decomposition strategy within 3DS, quantifying data difficulty through three metrics: *Instruction Understanding*, *Response Confidence*, and *Response Correctness*, ensuring fine-grained data difficulty quantification in the domain adaptation process.
- Our extensive experiments on Chinese medical datasets demonstrate that 3DS outperforms existing methods, significantly boosting LLMs performance in the medical domain. 3DS has also been successfully deployed in real-world medical applications (details omitted for anonymity).

- We have open-sourced a carefully curated Chinese medical dataset, including medical dialogues and domain-specific instructions, to support the fine-tuning of LLMs in healthcare.

## 2 RELATED WORK

### 2.1 DATA SELECTION FOR LLM

Data selection for LLM training has been explored through various approaches. Some works (Das & Khetan, 2023) utilize statistical clustering or core-set selection techniques to identify diverse and representative subsets, yet they neglect data quality and may incorporate noisy samples that hinder model training. To address quality concerns, some works leverage external models like proprietary LLMs (Chen et al., 2023a; Liu et al., 2023; Wettig et al., 2024) or reward models (Du et al., 2023) to evaluate and select high-quality training data. However, due to distribution differences and preference gaps between external models and the model to be trained, the selected data may not be beneficial for the model to be trained, leading to limited performance gains. Another line of research leverages information produced by the model to be trained, such as perplexity (Marion et al., 2023), gradients(Xia et al., 2024) and derived metrics like data learnability (Zhou et al., 2023) and instruction following difficulty (Li et al., 2024b;a). While these metrics provide more direct insights into the model's current understanding of data, they typically offer only coarse measures of data difficulty, failing to capture different aspects of data complexity or account for the model's generation behavior, leading to suboptimal selection. While these methods share similar challenges, insights and approaches with active learning methods Yoo & Kweon (2019); Karamcheti et al. (2021); Mindermann et al. (2022),their application scenarios and workflows are distinct. In this work, we focus exclusively on data selection tailored to the unique challenges of training LLMs. We note that existing data selection methods for LLMs are predominantly tailored for pre-training, general fine-tuning (transforming a base model into a chat model), or targeted for specific downstream tasks. There remains a significant absence in data selection for domain adaptation fine-tuning, where unique challenges lies in selecting data that effectively enhances the model's diverse domain abilities. To bridge this gap and overcome the limitations of current methods, our work introduces a novel data selection framework for domain adaptation and provides a more fine-grained analysis of data difficulty.

### 2.2 DATA LEARNABILITY IN LLM SFT

LLMs encounter significant challenges when learning unfamiliar or complex knowledge during supervised fine-tuning, particularly when the data was not encountered during pre-training, which can impede domain adaptation. Gekhman et al. (2024) found that models acquire new factual knowledge slowly during SFT, especially when the information diverges from their pre-existing understanding, leading to a higher risk of hallucinations. Ren et al. (2024) further show that when the knowledge introduced during Instruction Fine-tuning significantly differs from what was learned in pre-training, the model struggles to integrate it, causing performance degradation. This highlights the difficulty models face in using pre-training knowledge to understand new concepts. Kang et al. (2024) also emphasize that unfamiliar examples during fine-tuning increase the likelihood of hallucinations, suggesting that high-difficulty data can destabilize the model and negatively impact its ability to adapt to new domains. Together, these findings underscore the risks associated with fine-tuning on excessively difficult data, which can undermine model performance in domain-specific tasks.

## 3 METHODOLOGY

### 3.1 TASK FORMULATION

We define our task as Data Selection for Domain Adaptation, which focuses on selecting an optimal subset of domain-specific fine-tuning data to maximize an LLM's target domain performance. Given an initial LLM $M_\theta$ with parameter $\theta$ that has undergone pre-training and general instruction fine-tuning, e.g., LLaMA-chat, domain adaption aims to adapt the model to a specific target domain through continual fine-tuning using domain-specific instruction tuning data. Let $\mathcal{X}$ denote the full domain instruction fine-tuning dataset containing samples $x =< Q, A >$ with $Q = \{q_1, q_2, \ldots, q_m\}$ representing the instruction and $A = \{a_1, a_2, \ldots, a_n\}$ the response. Given a fixed budget $k$, the goal

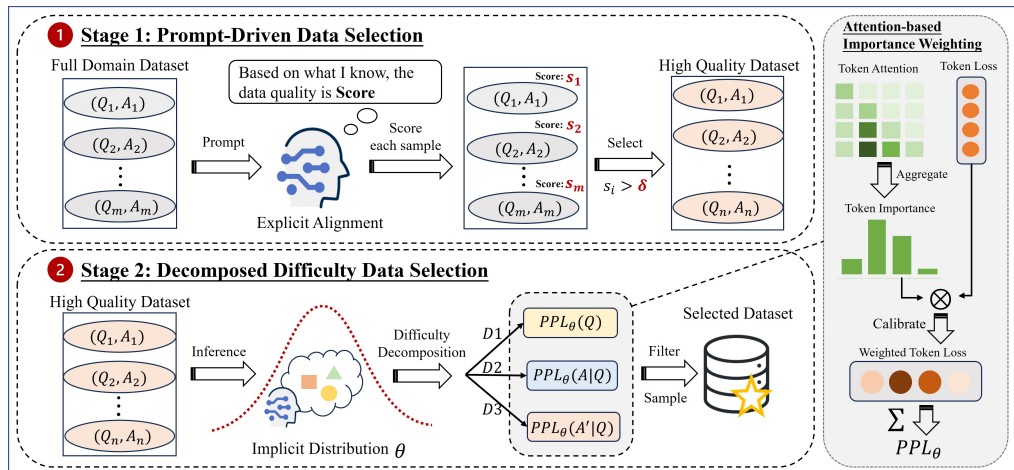

Figure 1: 3DS framework. **Stage 1** Prompt-Driven Data Selection select high-quality data via explicitly aligning data with the target LLM. **Stage 2** Decomposed Difficulty Data Selection decomposes data difficulty via modeling the LLM's implicit distribution and filter the dataset. Attention-based importance weighting mechanism calibrates difficulty calculation.

of data selection is to select a subset $\mathcal{S} \subseteq \mathcal{X}$ of size $k$ such that the model fine-tuned on $\mathcal{S}$ from $M_\theta$, achieves optimal performance in the target domain.

## 3.2 PROMPT-DRIVEN DATA SELECTION VIA EXPLICIT ALIGNMENT

The first stage of our framework is to select high-quality data that closely aligns with the inherent knowledge and preferences of the model to be trained. Unlike existing methods that rely on external reward models or proprietary LLMs to score data quality, which often result in suboptimal outcomes due to distributional mismatches and knowledge gaps, our approach directly uses the model itself for data evaluation. As illustrated in Figure 1, we leverage a carefully crafted prompt, detailed in Appendix A, to instruct the model to explicitly rate data quality based on its understanding. After obtaining the model-generated scores, samples with scores exceeding a predefined threshold $\delta$ are retained for further selection. By utilizing this prompt-driven alignment approach based on explicit model generation, our framework effectively reduces the gap between the training data and the model's inherent preferences, filtering out possible noise from low-quality or misaligned data.

## 3.3 DECOMPOSED DIFFICULTY DATA SELECTION VIA IMPLICIT DISTRIBUTION MODELING

The second stage of our framework is to analyze data difficulty via implicit distribution modeling of the model to be trained, thereby selecting data with moderate difficulty that best aligns with the model's learning capacity, to facilitate efficient domain adaptation. To achieve this, our Decomposed Difficulty Data Selection employs a fine-grained evaluation of data difficulty.

Inspired by the general problem-solving process Polya & Pólya (2014); OECD (2014)—understanding the problem, assessing confidence in the solution, and finally providing the answer—we decompose data difficulty into three key components that reflect the model's understanding: (1) **Instruction Understanding Difficulty** measures whether the model comprehends the given instruction. (2) **Response Confidence Difficulty** measures the model's ability to provide a confident and deterministic response based on the instruction. (3) **Response Correctness Difficulty** measures whether the model can generate a response that accurately matches the reference answer. In addition, we incorporate an **attention-based importance weighting mechanism** that calibrates difficulty by accounting for the varying semantic significance of tokens in the output, to ensure a more precise evaluation of response-related difficulties. Next, we will delve into the quantification of the decomposed difficulties and introduce the selection strategy.

**Instruction Understanding Difficulty** Challenging data often come with complex instructions, especially in specialized domains like healthcare, where instructions may contain intricate medical

terminologies. Accurately capturing how well a model understands such instructions is crucial, as a lack of comprehension indicates higher data complexity. To capture this aspect of data difficulty, we introduce Instruction Understanding Difficulty. Previous research (Gonen et al., 2023) has shown that a model's perplexity serves as an effective indicator of its familiarity with a prompt, where lower prompt perplexity correlates with better comprehension and performance. Building on this insight, we further recognize that perplexity inherently captures the predictive uncertainty from model's distribution. Consequently, we employ perplexity as a measure to quantify data difficulty from the model's perspective. Formally, for a model $M_\theta$, given a data sample $x = <Q, A>$ with the instruction $Q = \{q_1, q_2, \ldots q_m\}$, we define its Instruction Understanding Difficulty as:

$$\text{D1}_\theta(x) = \text{PPL}_\theta(Q) = \exp\left(-\frac{1}{m}\sum_{i=1}^{m}\log P_\theta(q_i|q_1, q_2, \ldots, q_{i-1})\right) \quad (1)$$

where $P_\theta(q_i|q_1, q_2, \ldots, q_{i-1})$ represents the probability model $M_\theta$ generates the $i$-th token in the instruction $Q$ given the preceding tokens. A higher perplexity value indicates greater difficulty for the model to comprehend the instruction.

**Response Confidence Difficulty** When encountering challenging data, the model often struggles to provide a confident response. This uncertainty arises from its inability to handle the task and determine the most appropriate response, similar to human students **?**, which indicates a high data difficulty. To quantify this difficulty, we introduce Response Confidence Difficulty, measured by the model's conditional perplexity when generating a response based on the instruction. Formally, for a model $M_\theta$, given a data sample $x = <Q, A>$ where $Q$ is the instruction and $A' = \{a'_1, a'_2 \ldots, a'_{n'}\}$ is the model-generated response based on $Q$, we define its Response Confidence Difficulty as:

$$\text{D2}_\theta(x) = \text{PPL}_\theta(A'|Q) = \exp\left(-\frac{1}{n'}\sum_{j=1}^{n'}\log P_\theta(a'_j|a'_1, a'_2, \ldots, a'_{j-1}, Q)\right) \quad (2)$$

where higher conditional perplexity indicates higher uncertainty in model's distribution and greater difficulty for the model to provide a confident answer.

**Response Correctness Difficulty** In instruction fine-tuning datasets that provide ground truths for given instructions, it is essential to assess the model's ability to generate accurate responses to assess data difficulty. We introduce Response Correctness Difficulty, measured by the model's conditional perplexity when generating the reference answer $A = \{a_1, a_2 \ldots, a_n\}$ based on the instruction $Q$.

$$\text{D3}_\theta(x) = \text{PPL}_\theta(A|Q) = \exp\left(-\frac{1}{n}\sum_{j=1}^{n}\log P_\theta(a_j|a_1, a_2, \ldots, a_{j-1}, Q)\right) \quad (3)$$

A higher conditional perplexity value reflects greater difficulty in producing the correct response, indicating that the data point is more challenging for the model.

**Attention-based importance weighting mechanism** Both Response Confidence Difficulty and Response Correctness Difficulty rely on evaluating the uncertainty inherent in the model's generation process. While conditional perplexity serves as a common method for uncertainty estimation, it treats all tokens within a response equally, disregarding their varying semantic importance. While key tokens significantly influence the meaning and correctness of a response, less important tokens like conjunctions or prepositions, may exhibit high uncertainty without substantially influencing the semantics. This can lead to skewed uncertainty estimates and inaccurate data difficulty assessments. To address this issue, inspired by Su et al. (2024), we introduce an attention-based importance weighting mechanism that adjusts perplexity-based measurements by weighting tokens according to their semantic importance. We argue that critical tokens are those playing a pivotal role in guiding the model's subsequent generation. Therefore, we derive importance scores from the model's internal attention mechanism. Specifically, for a token sequence $s = \{t_1, t_2, \ldots, t_i, \ldots, t_n\}$, when a transformer-based LLM generates token $t_j(i < j)$, it computes the attention weight $A_{ji}$ by applying a softmax function to the dot product of the query vector $\boldsymbol{q}_j$ and the key vector $\boldsymbol{k}_i$:

$$A_{ji} = (\frac{\boldsymbol{q}_j \cdot \boldsymbol{k}_i}{\sqrt{d_k}}) \quad (4)$$

where $d_k$ is the dimension of $k_i$. The attention weight $A_{ji}$ represents the attention the model pays to token $t_i$ when generating token $t_j$, reflecting the importance of token $t_i$. We define the importance score of token $t_i$ as the aggregated attention weight it receives from all subsequent tokens:

$$\mathrm{I}(t_i) = \underset{j > i}{\texttt{Aggregate}} \ (A_{ji}) \tag{5}$$

The aggregation function can either be the average (mean) of the maximum (max) value of all subsequent token scores. Using this attention-based importance score, we refine the calculations of Response Confidence Difficulty and Response Correctness Difficulty as follows:

$$\texttt{Atten-D2}_\theta(x) = \texttt{weightedPPL}_\theta(A'|Q)$$

$$= \exp\left(-\frac{\sum_{j=1}^{n'} \mathrm{I}(t_j) \cdot \log P_\theta(a'_j | a'_1, a'_2, \ldots, a'_{j-1}, Q)}{\sum_{j=1}^{n'} \mathrm{I}(t_j)}\right) \tag{6}$$

$$\texttt{Atten-D3}_\theta(x) = \texttt{weightedPPL}_\theta(A|Q)$$

$$= \exp\left(-\frac{\sum_{j=1}^{n} \mathrm{I}(t_j) \cdot \log P_\theta(a_j | a_1, a_2, \ldots, a_{j-1}, Q)}{\sum_{j=1}^{n} \mathrm{I}(t_j)}\right) \tag{7}$$

By integrating the attention-based importance weights, this mechanism ensures that tokens crucial for semantic correctness and clarity are prioritized, offering a more accurate estimation of model uncertainty and data difficulty.

**Selection Strategy based on Decomposed Difficulty** Based on the decomposed data difficulties, the selection algorithm first identifies samples whose difficulty metrics fall within a predefined middle range, filtering out either trivially easy or overly complex data, focusing on moderately challenging samples that matches model's learning capabilities. Once this subset is identified, we apply K-Center sampling based on instruction embeddings to enhance data diversity, reducing the risk of overfitting on highly similar samples. Details about K-Center sampling process are introduced in Appendix C.

---

**Algorithm 1:** Model-Driven Data Selection Framework

---

**Input:** Full dataset $\mathcal{X}$, model $M$, scoring threshold $\theta$, difficulty calculation functions
      D1, D2, D3, percentage thresholds $p_1, p_2, p_3$, sampling budget $k$
**Output:** Selected data subset $\mathcal{S}$
**Stage 1: Prompt-Driven Data Selection**
Initialize $\mathcal{X}_1 \leftarrow \emptyset$
**foreach** $x \in \mathcal{X}$ **do**
    Get score $s_x \leftarrow M(\text{prompt}, x)$
    **if** $s_x \geq \theta$ **then**
      | Add $x$ to $\mathcal{X}_1$
    **end**
**end**
**Stage 2: Decomposed Difficulty Data Selection**
Initialize $\mathcal{S} \leftarrow \emptyset$
Compute $\mathrm{D1}(x), \mathrm{D2}(x), \mathrm{D3}(x)$ for all $x \in \mathcal{X}_1$
Set $\tau_1, \tau_2, \tau_3$ based on percentiles $p_1, p_2, p_3$ of D1, D2, D3
**foreach** $x \in \mathcal{X}_1$ **do**
    **if** $\tau_1^{low} \leq D1(x) \leq \tau_1^{high}$ **and** $\tau_2^{low} \leq D2(x) \leq \tau_2^{high}$ **and** $\tau_3^{low} \leq D3(x) \leq \tau_3^{high}$ **then**
      | Add $x$ to intermediate set $\mathcal{S}_{\text{mid}}$
    **end**
**end**
Apply K-Center sampling on $\mathcal{S}_{\text{mid}}$ to select $k$ diverse data points
Return final selected subset $\mathcal{S}$

---

## 3.4 MODEL-DRIVEN DATA SELECTION FRAMEWORK

The overall architecture of our model-centric data selection framework is illustrated in Figure 1. The pseudo codes of the complete selection process are shown above.

## 4 EXPERIMENTS

### 4.1 EXPERIMENTAL SETUP

**Training dataset.** For medical domain adaptation, we construct a comprehensive medical instruction fine-tuning dataset of diversity and abundance. The dataset comprises over 1.9 million samples, with its statistics provided in Table 1. The details of data construction are introduced in Appendix B. We will release this complete training dataset to support further research.

| Dataset | Size (K) |
|---|---|
| medtalk_singleround | 177 |
| medknowledge_KG | 796 |
| medknowledge_webqa | 360 |
| medtask_promptcblue | 82 |
| qa_website | 490 |
| **Total** | **1905** |

Table 1: Training Dataset Statistics

| Dataset | Type | Size |
|---|---|---|
| CMB-Exam | multiple-choice | 11200 |
| MMCU-medical | multiple-choice | 2819 |
| CMB-Clin | open Q&A | 208 |

Table 2: Test Dataset Statistics

**Evaluation datasets.** We assess fine-tuned models on diverse medical test datasets, including two multi-task, multiple-choice datasets: MMCU-Medical (Zeng, 2023) and CMB-Exam (Wang et al., 2023c), and an open Q&A dataset, CMB-Clin (Wang et al., 2023c), with data statistics provided in Table 2. MMCU-Medical and CMB-Exam, consisting of medical exam questions using accuracy as the metric, assess the models' abilities to reason and apply medical knowledge. CMB-clin, comprising of patient record analysis tasks, assesses the model's ability to perform complex medical analysis. The metrics are BLEU-1, BLEU-4 and ROUGE, detailed in Appendix F. Together, these datasets provide a comprehensive evaluation of the model's proficiency in the medical domain.

**Models.** To validate the scalability and generalization ability of our data selection framework, we conduct experiments across chat models with different model architectures and parameter sizes, specifically `Baichuan2-7B-Chat`, `Baichuan2-13B-Chat` (Yang et al., 2023a), and `Qwen1.5-7B-Chat` (Bai et al., 2023).

**Baselines.** We compare 3DS against a series of LLM fine-tuning data selection strategies. (1)***Base*** directly tests the chat model without further fine-tuning. (2) ***Random Selection*** randomly selects samples. (3) ***IFD (Instruction-Following Difficulty)*** (Li et al., 2024a;b) designs a difficulty metric called instruction following difficulty based on the ground truth loss with or without the input instruction. (4)***MoDS (Model-oriented Data Selection)*** (Du et al., 2023) filters high-quality data via a reward model, and select data necessary for model learning through a two-stage training and inference process. (5)***LESS*** (Xia et al., 2024) searches for training samples similar to the target task examples through low-rank gradient similarity. The implementation details of the baselines are introduced in Appendix D.

**Implementations.** We fine-tune the models using the full training dataset, as well as subsets selected by our selection framework and the aforementioned baselines. For our method and all baselines, the training data budget is 5K samples. Models are fine-tuned using LoRA(Hu et al., 2021), with a learning rate of 5e-5 and a batch size of 64 for 1 epoch. Within our selection framework, the model-centric quality filtering stage retains data samples with a quality score exceeding 90. In the subsequent decomposed difficulty selection stage, we determine the difficulty thresholds through experiments on CMB hold-out validation set. Specifically, for `Baichuan2-7B-Chat`, the thresholds are set to 10% and 60%; for `Baichuan2-13B-Chat`, 15% and 65%; and for `Qwen1.5-7B-Chat`, 25% and 75%. More details about hyperparameters are introduced in Appendix E

### 4.2 MAIN RESULTS

Experiment results are shown in Table 3 and Table 4. We summarize our findings below.

**Data selection is necessary for LLM domain adaptation fine-tuning.** We observe that fine-tuning LLMs with the full 1.9 million dataset (Full-SFT) leads to drastic performance drops. This suggests that domain datasets directly collected from the internet contains noisy samples that hinder model learning, highlighting the necessity of data selection.

Table 3: Performance comparison (%) on *CMB-Exam*, *MMCU-Medical* of EM score. The best performance is highlighted in **bold**, and the second-best performance is underlined. Performance gains are measured against the base model.

| Method | LLM Turbo | Baichuan2-7B-Chat | | Baichuan2-13B-Chat | | Qwen1.5-7B-Chat | |
|---|---|---|---|---|---|---|---|
| | Dataset | CMB-Exam | MMCU-Medical | CMB-Exam | MMCU-Medical | CMB-Exam | MMCU-Medical |
| Baselines | Base | 24.50 | 21.67 | 46.67 | 47.11 | 59.80 | 64.24 |
| | Full-Sft | 21.53 | 22.49 | 40.38 | 37.90 | 48.05 | 47.53 |
| | Random | 23.02 | 23.13 | 44.07 | 47.61 | 61.81 | 65.10 |
| | MoDS | 24.90 | 23.48 | 47.25 | 50.37 | 61.09 | 64.67 |
| | IFD | 28.02 | 25.43 | 46.44 | 50.08 | **62.06** | 65.37 |
| | LESS | 25.30 | 23.84 | 45.79 | 51.01 | 60.74 | 64.85 |
| Ours | 3DS-$MeanAtten$ | 31.84 | 29.37 | **47.37** | 51.08 | 61.96 | **66.09** |
| | 3DS-$MaxAtten$ | 31.89 | 29.23 | 47.10 | 50.69 | 61.97 | 66.02 |
| | 3DS-$NoAtten$ | **32.05** | **29.51** | 47.10 | 50.19 | 61.79 | 65.84 |
| *Performance Gain ↑ | | 7.55 | 7.84 | 0.70 | 3.97 | 2.16 | 1.85 |
| Ablations | 3DS (w/o D1) | 30.30 | 27.81 | 47.35 | 50.59 | 61.47 | 65.80 |
| | 3DS (w/o D2) | 30.74 | 28.02 | 47.34 | 47.18 | 62.00 | 66.05 |
| | 3DS (w/o D3) | 31.22 | 28.80 | 47.07 | 50.59 | 61.64 | 65.73 |
| | 3DS (only D1) | 30.95 | 28.84 | 47.20 | **51.22** | 61.51 | 65.73 |

Table 4: Performance comparison (%) on *CMB-Clin*. The best performance is highlighted in **bold**, and the second-best performance is underlined. Performance gains are measured against the base model.

| Method | LLM Turbo | Baichuan2-7B-Chat | | | Baichuan2-13B-Chat | | | Qwen-1.5-7B-Chat | | |
|---|---|---|---|---|---|---|---|---|---|---|
| | Metric | BLEU-1 | BLEU-4 | ROUGE | BLEU-1 | BLEU-4 | ROUGE | BLEU-1 | BLEU-4 | ROUGE |
| Baselines | Base | 13.37 | 25.94 | 15.49 | 11.15 | 21.02 | 14.08 | 16.17 | 32.03 | 16.31 |
| | Full-Sft | 7.85 | 18.65 | 10.76 | 7.19 | 16.33 | 11.70 | 6.68 | 16.61 | 9.62 |
| | Random | 17.66 | 40.45 | 19.84 | 12.14 | 25.95 | 14.75 | 16.09 | 34.45 | 16.19 |
| | MoDS | 23.01 | 56.41 | 26.47 | 22.43 | 51.02 | 22.85 | 17.61 | 39.19 | 19.93 |
| | IFD | 22.80 | 60.59 | 29.83 | 21.44 | 51.73 | 24.94 | 19.24 | 43.10 | 21.08 |
| | LESS | 23.20 | 58.52 | 28.22 | 13.27 | 29.20 | 16.40 | 17.48 | 38.88 | 17.58 |
| Ours | 3DS-$MeanAtten$ | 22.61 | **64.57** | **32.11** | **24.15** | **63.51** | **31.50** | 24.40 | 60.32 | 28.07 |
| | 3DS-$MaxAtten$ | **23.94** | 63.58 | 31.48 | 23.49 | 61.95 | 30.22 | 24.58 | 60.47 | **28.23** |
| | 3DS-$NoAtten$ | 22.41 | 61.37 | 29.99 | 22.58 | 61.44 | 29.58 | **25.62** | **61.52** | 27.69 |
| *Performance Gain ↑ | | 10.57 | 38.63 | 15.99 | 13.00 | 42.49 | 17.42 | 9.45 | 29.49 | 11.92 |
| Ablations | 3DS (w/o D1) | 23.68 | 61.02 | 29.53 | 22.55 | 51.75 | 23.99 | 24.14 | 55.12 | 24.68 |
| | 3DS (w/o D2) | 22.96 | 61.35 | 30.46 | 22.22 | 52.06 | 23.54 | 20.48 | 49.59 | 23.84 |
| | 3DS (w/o D3) | 23.26 | 62.00 | 29.92 | 20.86 | 49.40 | 23.08 | 22.27 | 50.18 | 23.83 |
| | 3DS (only D1) | 22.89 | 61.76 | 30.58 | 22.09 | 52.01 | 23.91 | 21.92 | 51.48 | 26.16 |

3**DS effectively enhances LLM's diverse domain abilities, significantly outperforming baselines.** Baseline LESS, which focuses on enhancing model's targeted ability on a specific down-stream task, proves to be ineffective for domain adaptation where diverse abilities needs improvement. This approach leads to performance degradation on CMB-Exam for `Baichuan2-13B-Chat` and underperforms random sampling for `Qwen1.5-7B-Chat`. Similarly, MoDs fails to surpass random sampling for `Qwen1.5-7B-Chat` on CMB-Exam and MMCU-medical benchmarks, indicating that relying solely on external preferences without considering the distribution of the model to be trained are insufficient for enhancing domain-specific capabilities, especially for models already equipped with certain degree of domain knowledge. Among the baselines, IFD shows relative strong results due to its consideration of data difficulty, which aids in identifying beneficial samples that contribute to model learning. However, its instruction following difficulty is not comprehensive and the resulting performance improvements are marginal across tasks, even underperforming the base model on CMB-Exam for `Baichuan2-13B-Chat`. In contrast, our 3DS is the only method that consistently outperforms both the base model and random sampling across all benchmarks, bringing substantial performance gains to models of varying architectures and sizes. On medical exam datasets, our method improves model accuracy by up to 7.55% and 7.84% and surpass the best baselines by over 5.29% in accuracy. On the open Q&A CMB-clin, 3DS significantly outperforms all baselines by a large margin, with the fine-tuned model exhibiting superior medical analysis ability. These results validate that our proposed selection framework, which conducts data selection from a model-centric perspective and employs a fine-grained measurement of data difficulty through decomposition, consistently identifies effective training samples for LLM domain adaptation, universally enhancing their diverse domain abilities.

For CMB-Clin, we randomly sample 100 answers from models fine-tuned with different data selection methods and conduct a pair-wise evaluation using GPT-4 as the judge. The evaluation prompt

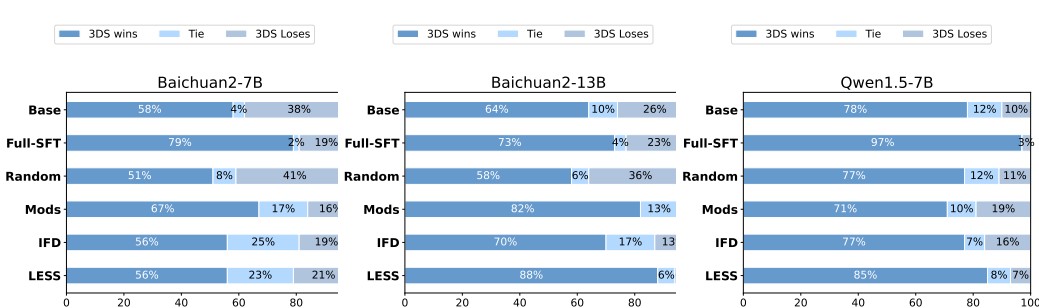

Figure 2: GPT4 judgement of CMB-Clin.

can be found in Appendix I. Results shown in Figure 2 further validate the superiority of our method. 3DS exhibits substantially higher win rates compared to all other baselines, achieving 67%, 82% and 71% win rates against MoDS, 56%, 70% and 77% against IFD, and 56%, 88% and 85% against LESS, for the three models respectively. This evaluation provides qualitative evidence that our method not only excels in quantitative metrics but also delivers more clinically accurate outputs.

**3DS exhibits strong generalization ability and scalability.** 3DS's consistent performance gains across various models and datasets highlight its great generalization ability to adapt to different models and domain tasks. Notably, on the CMB-Clin dataset, while all models benefit from our data selection strategy, the largest improvements are seen on the largest model, `Baichuan2-13B-Chat`. In Figure 2, the larger and stronger models `Baichuan2-13B-Chat` and `Qwen1.5-7B-Chat` also show generally higher win rates compared to `Baichuan2-7B-Chat`. These results validate that 3DS not only generalises well but also scales effectively with more capable models.

### 4.3 ABLATION STUDIES

To validate the effectiveness of each difficulty metric in our decomposed difficulties, we conduct ablation studies by removing each of the three metrics—Instruction Understanding Difficulty, Response Confidence Difficulty, and Response Correctness Difficulty. As shown in Table 3 and Table 4, in general, removing any single component result in noticeable performance drops on some evaluation metrics for all three models, indicating a decline in certain aspects of the model's medical domain abilities. For instance, the exclusion of Response Confidence Difficulty leads to a noticeable decrease in the performance of both `Baichuan2-7B-Chat` and `Baichuan2-13B-Chat` across all evaluation metrics. Similarly, `Qwen-1.5-7B-chat`'s performance drops on *CMB-Clin*. These observations validate the necessity of each difficulty metric in identifying beneficial data samples for enhancing LLM's domain abilities. Overall, the combination of these difficulty metrics contributes to a more accurate data difficulty measurement, ensuring that selected data matches the model's learning capacity and optimally enhances its domain performance. More ablation studies considering data budgets and selection steps are introduced in Appendix G.

### 5 IMPACT OF DIFFICULTY THRESHOLDS

To further investigate the relationship between training data difficulty and model performance in medical domain adaptation fine-tuning, we conduct a sliding-window experiment to identify the optimal training data for each model. Using hyperparameter $\sigma$ to denote the chosen difficulty level, we vary $\sigma$ and select training samples within the range $\sigma - 25\%$ and $\sigma + 25\%$ on our proposed difficulty metrics. As shown in Figure 3, for each model, performance improves as the training data difficulty increases, reaching a peak before declining. Notably, the optimal difficulty range differs depending on the model's inherent capability. For instance, `Baichuan2-7B-Chat` achieves its best performance when trained on data within relatively lower difficulty range of 10%-60%. For more powerful models like `Baichuan2-13B-Chat` and `Qwen1.5-7B-Chat`, the optimal ranges are 15%-65% and 25%-75% respectively, indicating that more capable models benefit from data of higher complexity. These findings further highlight the importance of selecting data that aligns with the model's capability. Training less capable models on excessively difficult data may overwhelm them, resulting in suboptimal performance, whereas models with stronger domain-specific knowledge require

more challenging domain data to enhance their abilities. This insight provides a valuable guideline for optimizing the fine-tuning process of LLMs for domain adaptation, and our difficulty metrics prove to be effective measures of data complexity.

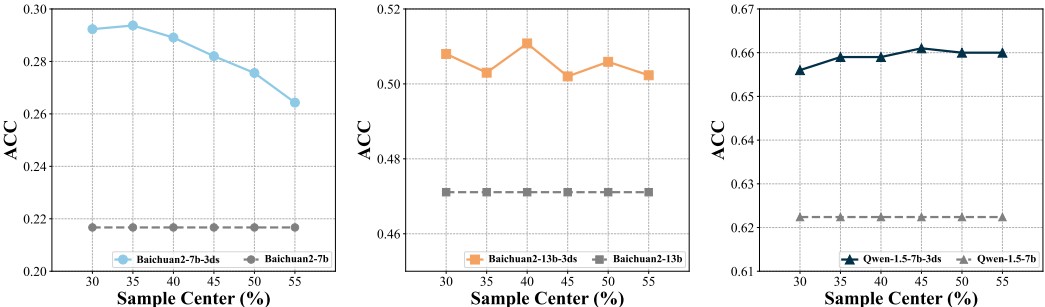

Figure 3: Impact of Difficulty Thresholds on Model Performance: The figure illustrates how varying difficulty thresholds of selection affect the accuracy (ACC) of models. The results are shown for `Baichuan2-7b-chat`, `Baichuan2-13b-chat`, and `Qwen-1.5-7b-chat`, across different difficulty sample centers (percentages).

## 6 CONCLUSION

In this paper, we introduce a two-stage model-centric data selection framework for LLM domain adaptation fine-tuning. The first stage performs a prompt-driven selection strategy to explicitly align with the model's preferences. The second stage selects data via data difficulty decomposition. By incorporating Instruction Understanding, Response Confidence, and Response Correctness difficulties, alongside an attention-based importance weighting mechanism, our method effectively captures the model's implicit distribution and selects data that matches the its learning capacity. Experimental results across multiple medical tasks demonstrate significant performance gains, validating the effectiveness of our selection framework. Our approach highlights the effectiveness of model-driven data selection, offering a path toward more efficient LLM domain adaptation training. Future work will explore extending this framework to other domains and refining the training procedure based on difficulty metrics for broader LLM applications.

## 7 LIMITATIONS

Due to time and resource constraints, we have only validated our method in the medical domain. While our data selection framework is domain-agnostic and adaptable to other fields, further experiments in other domains are needed to fully verify its generalization. Since the selection process requires the model to perform inference on the training data, it involves certain computational costs. This additional inference step may increase computational overhead, especially when working with very large datasets. Our framework performs data selection prior to LLM fine-tuning. Considering that the model's evaluation of data difficulty may evolve during training, future research should explore dynamic selection that adapts to the model's changing state. Additionally, data filtered out is currently discarded. Future work should consider integrating mechanisms such as human-in-the-loop validation or strategies to recover potentially relevant and valuable data from the discarded pool. Finally, considerations for social bias and fairness issues are discussed in Appendix J.

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

## A    DATA QUALITY EVALUATION PROMPT

In the first stage of our proposed data selection framework, we carefully craft a prompt to instruct the current model to evaluate the training set and filter out noisy data samples based on its internal knowledge. Inspired by existing works Chen et al. (2024); Wang et al. (2023c); Liu et al. (2023), the model is asked to assess data quality across five dimensions: Instruction Complexity, Response Relevance, Response Thoroughness, Response Logic and Knowledge Richness. We provide the model with detailed scoring guidelines. The specific prompt used in this process is shown below.

---

**Quality Evaluation Prompt in Stage 1**

You are an AI assistant with medical expertise. Your task is to objectively assess the quality of the medical dialogue between the user and assistant based on your knowledge, and provide a score. The data may consist of single or multi-turn dialogues. You should evaluate based on the complexity of the question, relevance of the response, thoroughness, logical coherence, and knowledge richness, and provide an overall score. Focus on medical-specific characteristics to ensure accuracy.

*[Evaluation Criteria]*

1. *Question Complexity*: Evaluate the complexity of the user's question. If the question requires deep understanding, reasoning, or medical knowledge, score above 80.

2. *Response Relevance*: Assess if the assistant's response is directly aligned with the question. Score above 80 for responses tightly related to the question.

3. *Response Thoroughness*: Check if the response thoroughly addresses the question with sufficient detail. A score above 80 reflects comprehensive answers.

4. *Response Logic*: Ensure the response follows clear reasoning and logic. A score above 80 reflects well-structured reasoning.

5. *Knowledge Richness*: Determine whether the response demonstrates rich, specialized medical knowledge. A score above 80 indicates depth and accuracy.

*[Scoring Guidelines]*

[80-100]: Excellent. High complexity, thoroughness, relevance, logic, and knowledge richness, meeting medical standards.

[60-79]: Good. Strong performance but with minor deficiencies in logic or knowledge.

[40-59]: Fair. Noticeable issues such as unclear logic or insufficient depth.

[20-39]: Poor. Fails to properly address the medical issue or lacks substance.

[0-19]: Very Poor. Lacks relevance, logic, or medical knowledge.

*[Start Conversation]*
Refer to the guidelines and score the following dialogue data based on the criteria. Follow the output format strictly:
{score:}
Dialogue:
`<qa_pairs>`
Output:

---

## B    DATASHEET FOR MEDICAL DOMAIN ADAPTATION FINE-TUNING DATASET

**What is the primary purpose of creating this dataset?**

This dataset was created to construct a large-scale medical domain instruction-following fine-tuning dataset. The primary purpose is to support the adaptation of large language models (LLMs) to the medical domain by providing diverse and comprehensive training instances. By integrating heterogeneous data sources, including doctor-patient dialogues, medical knowledge bases, and various medical tasks formulated into the instruction-output format, the dataset aims to enhance the ability of LLMs to perform effectively across a wide range of real-world medical scenarios. It is designed to address the unique challenges of the medical domain, such as specialized terminology, complex reasoning, and context-sensitive responses, thereby enabling LLMs to better meet the demands of healthcare applications.

**What are the specific components of the dataset, and how were they constructed or sourced?**

Our dataset integrates multiple open-sourced medical instruction fine-tuning datasets from diverse sources, along with doctor-patient dialogue data extracted from medical consultation websites and a variety of medical tasks reformulated into the instruction-output format, as detailed in Table 1. **Medtalk_singleround** originates from open-sourced doctor-patient question-and-answer datasets, including CMedQA2 (Zhang et al., 2018) and Health-Care-Magic[1]. **Medknowledge_KG** is built from the Online Medical Knowledge-Based Data in Huatuo26M (Li et al., 2023), which is derived from the extensive medical literature data provided by the Chinese Medical Association. **Medknowledge_webqa** includes knowledge-driven, open-ended question-and-answer pairs in the medical domain, sourced from (Wang et al., 2023b). **Medtask_promptcblue** combines the promptCBLUE dataset (Zhu et al., 2023b) with additional data converted into the instruction-output format from the CBLUE benchmark (Zhang et al., 2022). **QA_website** contains authentic doctor-patient dialogue data collected from the online platform of a collaborating hospital. Examples from these datasets are shown in Table 5.

**Are the data sources legal? How are privacy and ethical considerations addressed?**

The dataset is derived from carefully selected sources, including publicly available datasets and data crawled from the website of a collaborating hospital. Explicit permission was obtained from the collaborating hospital for the use of the crawled data, and all data have been anonymized to ensure that no personal information is exposed. Additionally, the hospital's website provides open-access data, complying with relevant legal and ethical standards. This ensures the legality and security of the data while addressing privacy and ethical concerns.

**What are the potential risks and limitations of this dataset?**

The dataset has certain inherent risks and limitations that should be acknowledged. First, as the data is collected from diverse sources, it may contain noise or inconsistencies, which could affect the quality and reliability of downstream applications. Additionally, since the dataset is derived from Chinese text corpora, including medical advice and Q&A exchanges, its content may be culturally and regionally specific, making it more suitable for East Asian populations. As a result, the medical recommendations and insights in the dataset may not generalize well to other demographic or cultural contexts.

To address these issues, users should carefully evaluate the dataset's suitability for their intended applications and, if necessary, consider adapting the data to align with broader use cases. Moreover, noise reduction and validation techniques can be employed to improve data quality and reliability in specific tasks.

**What is the usage case for this dataset?**

This dataset is primarily intended for instruction fine-tuning of large language models (LLMs), as already utilized in this study. Practitioners can use it to fine-tune LLMs to adapt to the medical domain, as well as to enhance its medical abilities in general fine-tuning. Additionally, the dataset may be useful for more specific tasks, such as fine-tuning for sub-tasks in the dataset.

---

[1] https://www.kaggle.com/datasets/gunman02/health-care-magic

**What is the distribution method and maintenance plan for this dataset?**

The dataset is distributed as an open-source resource at `https://drive.google.com/drive/folders/1SfrwQkDrQJ8i_EIqfc2Di0Xa5Y5pzY9H`, allowing researchers and developers to access and utilize it freely under the specified license. We are committed to the ongoing maintenance of the dataset. If any errors or inaccuracies are identified, particularly those related to medical knowledge, we will promptly update the dataset to correct such issues, removing erroneous data as necessary. Additionally, we will continue to provide updated documentation to ensure the dataset's effective use. While the dataset is stable at present, users are encouraged to provide feedback or suggest improvements, and we will consider updates based on user input or evolving needs in the field. This ensures that the dataset remains reliable and beneficial for the community.

## C    K-CENTER SAMPLING ALGORITHM

In our data selection framework, K-Center sampling is employed to ensure diversity within the selected instruction fine-tuning data. After filtering based on difficulty levels, we obtain an intermediate set $S_{\text{mid}}$, composed of data points within a moderate difficulty range. The K-Center sampling is then applied on $S_{\text{mid}}$. Specifically, the process works as follows:

1. Embedding Generation: For each data sample, the instruction part is encoded into an embedding using the LLM. We extract the last hidden states of the LLM and compute the average across all tokens in the sequence to form a fixed-size embedding vector. These embeddings represent the semantic content of the instruction.

2. K-Center Sampling: Using these embeddings, the K-Center sampling algorithm selects $k$ data points in a greedy manner. The goal is to maximize the minimum distance between any pair of selected data points, ensuring that the sampled data points are as distinct as possible. This promotes diversity in the selected dataset and minimizes the risk of overfitting to similar data points.

The pseudo codes of this greedy K-Center sampling process are shown below:

---

**Algorithm 2:** Greedy K-Center Sampling

---

**Input:** Intermediate set $S_{mid} = \{s_1, s_2, \ldots, s_n\}$, model $M$, data budget $k$
**Output:** Final selected set $\mathcal{S}$
**Step 1: Encode data in $S_{mid}$ using model $M$**;
**foreach** $s_i \in S_{mid}$ **do**
  | Encode $s$ using $M$ to obtain the embedding $e_s$ ;
**end**
**Step 2: Run K-Center greedy algorithm**;
Initialize $\mathcal{S} \leftarrow \emptyset$ ;
Initialize min_distances $\leftarrow \infty$ ;
**for** $i = 1$ **to** $k$ **do**
  | **if** $\mathcal{S} = \emptyset$ **then**
  | | Select $s_j \in S_{mid}$ randomly and add it to $\mathcal{S}$ ;
  | **else**
  | | $min\_distances_j = \min_{s_i \in \mathcal{S}} \|e_{s_j} - e_{s_i}\|_2, \quad \forall s_j \in S_{mid} \setminus \mathcal{S}$;
  | | Select $s^* = \arg\max_{s_j \in S_{mid} \setminus \mathcal{S}} min\_distances_j$;
  | | Add $s^*$ to $\mathcal{S}$;
  | **end**
**end**
**return** $\mathcal{S}$

---

## D    BASELINE IMPLEMENTATIONS

Due to differences in task settings and datasets, we re-implement the baselines using their publicly available codes. We adapt their data selection strategies to our domain adaptation task on the medical instruction fine-tuning dataset and models. The re-implementation details are as follows:

| | Medtalk_singleround | English translation |
|---|---|---|
| Question | 医生请问怀孕时母亲得了甲亢会遗传个孩子么？之前得过甲亢怀孕时又复发了但是没有吃药想知道宝宝会不会被遗传？ | Q: Doctor, can hyperthyroidism during pregnancy be inherited by the baby? Mine recurred during pregnancy, but I didn't take medication. Will my baby be affected? |
| Answer | 甲亢与遗传有一定的关系，但不是都会遗传的，不要担心。建议平时合理安排饮食，需要高热量高蛋白质高维生素和低碘的饮食。精神要放松。适当休息，避免重体力活。 | A: Hyperthyroidism has some genetic links but is not always inherited. Don't worry. Maintain a balanced diet rich in calories, protein, and vitamins, but low in iodine. Stay relaxed, rest adequately, and avoid strenuous activities. |
| | **Medknowledge_KG** | **English translation** |
| Question | 颜面部凹陷的手术治疗有些什么？ | Q: What surgical treatments are available for facial depression? |
| Answer | 自体颗粒脂肪移植；自体脂肪移植；自体脂肪干细胞移植；自体脂肪颗粒移植。 | A: Autologous micro-fat grafting, autologous fat transplantation, autologous stem cell-enriched fat grafting, and autologous fat particle transplantation. |
| | **Medknowledge_webqa** | **English translation** |
| Question | 请描述寄生虫病的治疗方法。 | Q: Please describe the treatment methods for parasitic diseases. |
| Answer | 以消灭寄生虫为主，根据虫种采用最有效的驱虫药物。在感染较重而寄主较衰弱时，可给予支持疗法，有外科并发症时应及时进行外科处理。 | A: The primary goal is to eliminate the parasites by using the most effective antiparasitic drugs based on the specific type of parasite. In cases of severe infection where the host is weakened, supportive therapy may be provided. Surgical intervention should be performed promptly if complications arise. |
| | **Medtask_promptcblue** | **English translation** |
| Question | 问诊对话的实体抽取：两个礼拜片感冒了，现在还有点咳嗽，会不会和这个有关？选项：具体的药物名称，医疗操作，医学检查检验，症状。 | Q: Entity extraction in diagnostic dialogues: "Caught a cold two weeks ago, still have a bit of a cough now—could it be related?" Options: specific medication names, medical procedures, medical tests and examinations, symptoms. |
| Answer | 上述句子中的实体包含：症状实体：感冒，咳嗽。 | A: The entities in the above sentence include: Symptom entities: cold, cough. |
| | **QA_website** | **English translation** |
| Question | 每天下午低烧三十六七℃，有时胸闷，没有咳嗽，盗汗，乏力的，有没有得肺结核的可能？ | Q: Low-grade fever of 36-37℃ every afternoon, occasional chest tightness, no cough, night sweats, or fatigue—could this indicate a possibility of tuberculosis? |
| Answer | 你这个体温其实从临床上来讲，不算是低烧，一般来讲，37度二以上才算是低热，所以说你这个跟集合的关系不是特别大的，你倒是可以看一下有没有病毒感染的可能，再一个，有没有新冠的问题？ | A: From a clinical perspective, this temperature doesn't qualify as a low-grade fever—typically, temperatures above 37.2℃ are considered low-grade. Therefore, its connection to tuberculosis is unlikely. However, you might want to check for the possibility of a viral infection or consider whether it could be related to COVID-19. |

Table 5: Examples For various type dataset

**(1) IFD:** Li et al. (2024a;b) The Instruction Following Difficulty (IFD) method begins by calculating the instruction-following difficulty scores for each data point through model forward propagation. Given that our full domain dataset consists of over 1.9 million samples, performing this step on the entire dataset would be computationally prohibitive. Therefore, we randomly sample

60k samples from the training set, an amount comparable to the dataset size used in our 3DS after stage 1. We compute IFD scores for this subset, and, following the recommendations in the original paper, select the samples with highest scores. The data budget is constrained to 5k samples, ensuring consistent with our main experimental setup.

**(2) MoDS:** Du et al. (2023) For the MoDS baseline, We follow the original paper's implementations, using the reward model `reward-model-deberta-v3-large-v2`[2] to score the full dataset. We then obtain samples with scores above 0.5, yielding a subset of 120k high-quality data samples. From this subset, we apply K-Center sampling to select 2k seed samples for model warm-up training. Subsequently, the trained model perform inference on the 120k high-quality subset, and these predictions are rescored using the same reward model. Data samples where model's generated answers score below 0 are deemed necessary and are combined with the seed samples. From this merged set, we randomly select 5k samples as the final training data, and train models from scratch on this final data.

**(3) LESS:** Xia et al. (2024) The LESS method involves constructing a gradient library based on the original data, which incurs significant computational costs, particularly for the large dataset like ours. Similarly, we sample 60k data points to compute the gradients. Unlike the original LESS method that targets specific downstream tasks and uses samples from the targeting dataset to construct a validation set, our domain adaptation scenario does not involve fixed downstream tasks. Therefore, we randomly selected an additional 100 samples from the training set as the validation set. Then we run the provided codes and select 5k training samples.

## E  HYPERPARAMETERS

Table 6: Performance of models on the hold-out set.

| Model | 5-55 | 10-60 | 15-65 | 20-70 | 25-75 | 30-80 |
|---|---|---|---|---|---|---|
| Baichuan2-7B | 28.22 | **28.99** | 27.67 | 28.22 | 24.18 | 24.25 |
| Baichuan2-13B | 37.58 | 36.83 | **37.81** | 37.63 | 38.58 | 37.65 |
| Qwen1.5-7B | 56.97 | 57.60 | 57.80 | 58.23 | **58.65** | 57.91 |

The difficulty thresholds in our experiments are determined based on model performance on a hold-out CMB-validation set composed of 280 samples provided in the CMB benchmark Wang et al. (2023c). As shown in Table 6, we select the optimal difficulty thresholds for each model based on their validation performance. Specifically, the resulting thresholds are 10% and 60% for `Baichuan2-7B-Chat`; 15% and 65% for `Baichuan2-13B-Chat`; and 25% and 75% for `Qwen1.5-7B-Chat`. All experiments are conducted on 8 NVIDIA H100 GPUs, with both training and inference performed using half-precision FP16 for efficiency. We employ the LoRA fine-tuning method, targeting all linear modules within the model, with a learning rate of $5 \times 10^{-5}$, a batch size of 64, and a single epoch of training. The learning rate is scheduled using a cosine decay scheduler with a warmup ratio of 0.1. The LoRA rank is set to 8, and the input sequence length is cut off at 1024 tokens. DeepSpeed Zero-3 is used to optimize distributed training. For instruction scoring, response generation, and training, we use templates corresponding to each model, implemented through the llamafactory project Zheng et al. (2024).

## F  EVALUATION METRICS

To evaluate the performance of LLMs on multi-task medical choice questions, we instruct the models to provide only the correct answer and adopt the widely-used metric, **Exact Match (EM)**, as recommended by prior work Zhu et al. (2021); Karpukhin et al. (2020). An answer is deemed correct under the EM metric if its form exactly matches all the correct answers listed in the ground truth. The EM score is computed as follows:

$$EM = \frac{\text{Number of Correctly Matched Answers}}{\text{Total Number of Answers}} \times 100\%.$$

---

[2]https://huggingface.co/OpenAssistant/reward-model-deberta-v3-large-v2

For open-domain medical Q&A tasks, we employ **ROUGE-R** Xu (2023); Jiang et al. (2024) and **Bilingual Evaluation Understudy (BLEU)** to assess the quality of the LLMs' responses.

**BLEU-N** Specifically, **BLEU-1** is used to measure answer precision, and **BLEU-4** evaluates answer fluency by considering higher-order n-gram consistency. **BLEU** evaluates the similarity of generated responses to the ground truth using the following formula:

$$\text{BLEU-N} = BP \cdot \exp\left(\frac{1}{N}\sum_{n=1}^{N} \log p_n\right),$$

where $p_n$ is the precision of $n$-grams, $BP$ is the Brevity Penalty, calculated as:

$$BP = \begin{cases} 1, & \text{if } c > r \\ \exp\left(1 - \frac{r}{c}\right), & \text{if } c \leq r \end{cases}.$$

Here $c$ is the length of the generated response, and $r$ is the length of the reference response.

**ROUGE-R** quantifies the recall of retrieved knowledge in the LLMs' responses, emphasizing their ability to comprehensively cover the information relevant to the query. For a generated response $R$ and a reference $G$, ROUGE-R is computed as:

$$\text{ROUGE-R} = \frac{|R \cap G|}{|G|},$$

where $|R \cap G|$ denotes the number of overlapping n-grams between the generated response and the reference, and $|G|$ is the total number of n-grams in the reference.

# G  SUPPLEMENTAL ABLATION STUDIES

## G.1  ABLATIONS ON SELECTION STAGES

Table 7: Performance comparisons (%) on *CMB-Exam*, *MMCU-Medical* of removing individual steps and collapsing stage 2 into stage 1 across different datasets and models. The best performance is highlighted in **bold**, and the second-best performance is underlined. The original method (3DS-Mean Attention) consistently outperforms the ablation variants.

| LLM Turbo | Baichuan2-7B-Chat | | Baichuan2-13B-Chat | | Qwen1.5-7B-Chat | |
|---|---|---|---|---|---|---|
| Dataset | CMB-Exam | MMCU-Medical | CMB-Exam | MMCU-Medical | CMB-Exam | MMCU-Medical |
| Without Stage 1 | 29.61 | 28.88 | 44.64 | 48.06 | 60.37 | 64.03 |
| Without Stage 2 | 29.41 | 27.03 | 47.09 | 50.83 | 61.59 | 65.91 |
| Stage 2 Collapsed into Stage 1 | 29.09 | 25.97 | 47.28 | 51.01 | 60.56 | 63.99 |
| 3DS-$MeanAtten$ | **31.84** | **29.37** | **47.37** | **51.08** | **61.96** | **66.09** |

Table 8: Performance comparison (%) on *CMB-Clin* of removing individual steps and collapsing stage 2 into stage 1 across different datasets and models. The best performance is highlighted in **bold**, and the second-best performance is underlined. The original method (3DS-Mean Attention) generally outperforms the ablation variants.

| LLM Turbo | Baichuan2-7B-Chat | | | Baichuan2-13B-Chat | | | Qwen-1.5-7B-Chat | | |
|---|---|---|---|---|---|---|---|---|---|
| Metric | BLEU-1 | BLEU-4 | ROUGE | BLEU-1 | BLEU-4 | ROUGE | BLEU-1 | BLEU-4 | ROUGE |
| Without Stage 1 | 17.01 | 38.52 | 19.39 | 14.13 | 29.60 | 16.19 | 15.50 | 31.94 | 15.88 |
| Without Stage 2 | 21.29 | 55.74 | 27.62 | 20.56 | 46.86 | 21.83 | 21.55 | 47.39 | 21.55 |
| Stage 2 Collapsed into Stage 1 | **22.71** | 60.13 | 29.46 | 21.48 | 50.16 | 22.69 | 21.73 | 52.27 | 23.41 |
| 3DS-$MeanAtten$ | 22.61 | **64.57** | **32.11** | **24.15** | **63.51** | **31.50** | **24.40** | **60.32** | **28.07** |

Table 9: Win-rates (%) of GPT-4 judgment on *CMB-Clin*, comparing 3DS-$MeanAttention$ with ablation variants.

| LLM Turbo | Baichuan2-7B-Chat | | | Baichuan2-13B-Chat | | | Qwen-1.5-7B-Chat | | |
|---|---|---|---|---|---|---|---|---|---|
| Metric | Win | Tie | Lose | Win | Tie | Lose | Win | Tie | Lose |
| vs Without Stage 1 | 65.5 | 12.5 | 22.0 | 66.5 | 9.0 | 24.5 | 70.5 | 3.0 | 26.5 |
| vs Without Stage 2 | 65.5 | 11.0 | 23.5 | 66.0 | 15.5 | 28.5 | 66.0 | 5.5 | 28.5 |
| vs Stage 2 Collapsed into Stage 1 | 62.0 | 9.5 | 28.5 | 63.5 | 18.0 | 18.5 | 54.5 | 2.5 | 43.0 |

Our proposed data selection framework is composed of two stages: 1. select high-quality data by prompting the model; 2. calculate decomposed data difficulties utilizing model perplexity. To evaluate the contributions of each stage, we investigate the impact of removing each stage and conduct a series of ablation experiments. The experiments include **(1) removing Stage 1**, where 70,000 samples are randomly sampled from the complete training dataset for subsequent difficulty calculation and filtering, and **(2) removing Stage 2**, where direct K-Center sampling is applied to the high-quality samples identified in stage 1' without difficulty filtering. Additionally, to further validate the necessity of decomposed difficulty calculation based on model perplexity, we test **(3) collapsing Stage 2 into Stage 1**, where the model is prompted to verbalize its assessments of the three data difficulties (Instruction Understanding Difficulty, Response Confidence Difficulty, Response Correctness Difficulty, with corresponding prompts shown below), bypassing the original difficulty calculation.

The results highlighted in Table 7, Table 8 and Table 9 show a consistent pattern: **each modification leads to a decrease in performance compared to the original method (3DS-Mean Attention), which consistently remains the best-performing approach across all models and all testing benchmarks.**

Removing Stage 1 leads to significant performance degradation, demonstrating the importance of quality control. Removing Stage 2 also results in performance declines, further emphasizing the necessity of selecting appropriately difficult data for effective model fine-tuning. When Stage 2 is collapsed into Stage 1 via additional difficulty prompts, performance also degrades. During experiments, we observed that the model struggles to provide fine-grained assessments of data difficulty, often generating coarse-grained scores such as 0.5, 0.8, and 1. This lack of granularity makes it challenging to identify nuanced differences in data difficulty. Furthermore, without knowing the exact capabilities of the model, we could not design in-context learning examples to guide finer-grained difficulty judgments. Filtering based on model-prompted difficulties typically results in 20k-30k samples from an initial pool of 60k-70k, whereas the perplexity-based difficulty calculation reduces the selection to fewer than 10k samples. This smaller, more targeted dataset aligns better with the desired moderate difficulty range, leading to improved fine-tuning performance.

Although the performance differences between the proposed method and the ablation variants are not very pronounced on CMB-Exam and MMCU-Medical, our method is notably the most consistent across different models. Other variants, despite performing well on one model, tend to show degradation on another. For instance, collapsing Stage 2 into Stage 1 results in relatively good performance on `Baichuan2-13B-Chat`, but performs poorly on `Baichuan2-7B-Chat`. In contrast, our method maintains steady good performance across different models, underscoring its robustness and reliability.

Furthermore, domain-adaptation aims to enhance the model's diverse abilities in the target domain, where models need not only to answer questions accurately but also to analyze and present content effectively. The results on the medical analysis task CMB-Clin shown in Table 8 clearly demonstrate that our method significantly outperforms the ablation variants, exhibiting superior medical analysis capabilities. While the multiple-choice results did not conclusively indicate which stage is most important, the analysis performance reveals a clear trend: **removing Stage 1 leads to the poorest performance, followed by removing Stage 2, and collapsing Stage 2 into Stage 1 achieves better results than both**. This pattern highlights the crucial role of quality control in determining the model's ability to provide coherent and high-quality answers. At the same time, difficulty filtering is also essential, as even the coarser-grained difficulty measurement by model verbalization yields better results than ignoring difficulty at all. This progressive improvement reinforces the importance of the filtering metrics we consider, showing that both quality and difficulty are vital for selecting beneficial data.

We also conduct GPT-4 judgment to compare the analysis generated by the original method and other alternatives. The win-rate results in Table 9 reinforce the superiority of our original approach. When comparing 3DS-$MeanAttention$ with ablation variants, the win-rate generally exceeds 60%, indicating a significant preference for the proposed method over its alternatives.

These results together indicate that both steps of the original algorithm are crucial for maximizing performance, and that the calculation method of data difficulties cannot yet be replaced by model-verbalized assessments.

> **Instruction Following Difficulty Prompt**
>
> Based on your existing knowledge, evaluate the difficulty of understanding the following instruction. The higher the complexity and ambiguity of the instruction, the more difficult it is for the model to understand. Please provide a score between 0 and 1, where a higher score indicates that the instruction is more difficult for you to understand.
>
> **Instruction to be evaluated:** {instruction}
>
> Please return a real number between 0 and 1, representing the difficulty of understanding the instruction. Only output the score, and do not output anything else.

> **Response Confidence Difficulty Prompt**
>
> Based on your existing knowledge, evaluate the difficulty of confidently and definitively providing the following evaluated response to the instruction. The more difficult it is to confidently provide this response, the higher the difficulty. Please provide a score between 0 and 1, where a higher score indicates greater difficulty in answering confidently.
>
> **Instruction:** {instruction}
> **Response to be evaluated:** {generated output}
>
> Please return a real number between 0 and 1, representing the difficulty of confidently providing the response to the instruction. Only output the score, and do not output anything else.

> **Response Correctness Difficulty Prompt**
>
> Based on the following instruction and the standard answer, evaluate the difficulty of providing the correct standard answer. If the instruction is complex or the answer requires high expertise, making it difficult to provide the correct answer, the difficulty will be higher. Please provide a score between 0 and 1, where a higher score indicates greater difficulty in providing the correct answer.
>
> **Instruction:** {instruction}
> **Standard Answer:** {output}
>
> Please return a real number between 0 and 1, representing the difficulty of providing the correct answer. Only output the score, and do not output anything else.

## G.2 COMPARISON WITH SELECTION VIA EXTERNAL LLM ANNOTATION

Table 10: Performance comparisons with models trained on data selected using Qwen2.5-7B as the quality evaluator

| Dataset | CMB-Exam | | MMCU-Medical | |
|---|---|---|---|---|
| Model | QwenRate | 3DS | QwenRate | 3DS |
| Baichuan2-7B | 28.55 | **31.84** | 25.90 | **29.37** |
| Baichuan2-13B | 46.82 | **47.37** | 50.55 | **51.08** |
| Qwen1.5-7B | 60.67 | **61.96** | 64.17 | **66.09** |

To further evaluate the effectiveness of our proposed 3DS, we conduct a comparison with data selection based on external LLM annotations. This experiment aims to investigate whether our method can match or surpass the performance of a costly external LLM-based approach in identifying beneficial data for model training, without incurring additional costs. In this experiment, we use Qwen2.5-72B, a state-of-the-art Chinese LLM, as the external data quality evaluator. The

evaluation process follows the same quality evaluation prompt used in our method and 5000 data points scoring of 85 or higher are selected and used to train the models. Table 10 presents the experimental results. Across all tested models and benchmarks, the models trained using our 3DS consistently outperform those trained on data selected by Qwen2.5-72B. The results demonstrate that our model-centric 3DS data selection approach effectively identifies beneficial data that leads to superior model performance compared to external LLM-based annotation. Importantly, our method achieves these results without incurring additional annotation costs, further validating the practicality of model-centric data selection. These findings underscore the potential of leveraging the model itself to guide data selection in a cost-effective and performance-optimized manner.

## G.3 Comparison with Existing Medical LLMs

Table 11: Performance comparisons with existing medical LLMs.

| Model | CMB-Exam | MMCU-Medical |
|---|---|---|
| Baichuan2-7B-3DS | 31.84 | 29.37 |
| Baichuan2-13B-3DS | 47.37 | 51.08 |
| Qwen1.5-7B-3DS | **61.96** | 66.09 |
| Meditron-7B | 11.20 | 12.16 |
| Huatuo-7B | 27.69 | 47.18 |
| Huatuo-34B | 59.54 | **66.10** |

To further validate the practical utility of our proposed 3DS framework, we conduct a comparison against existing medical LLMs. This experiment aimed to evaluate whether our approach can achieve competitive or superior performance compared to established medical LLMs, including open-source models MediTron Chen et al. (2023b) (7B version due to its similar size to Baichuan2-7B and Qwen1.5-7B), and state-of-the-art Chinese medical LLMs HuatuoGPT-II-7B, and HuatuoGPT-II-34B Chen et al. (2024). The results of the comparison are presented in Table 11. MediTron-7B, as an English-based LLM, demonstrates limited performance on Chinese medical benchmarks, significantly underperforming other models. Huatuo-7B shows strong results on MMCU-Medical, exceeding Baichuan2-7B-3DS, but falls short on the more complex and larger CMB-Exam. This suggests that while Huatuo-7B captures certain domain-specific information, it struggles with broader and more diverse tasks. Huatuo-34B, with nearly five times the size of Qwen1.5-7B, achieves comparable performance with Qwen1.5-7B-3DS. However, this comes with significantly higher computational and resource requirements.

It is worth noting that the performance of fine-tuned models is closely tied to the capability of the base model, so relative improvements achieved through domain-specific fine-tuning are more important than absolute performance. Still, the strong performance of models fine-tuned with 3DS validates its practical utility and efficiency for developing medical domain LLMs, paving ways for more building more powerful and advanced models in the future.

## G.4 Ablation on Data Budgets

Table 12: Performance comparison of models trained on different data budgets.

| Model | Dataset | 3k | 4k | 5k | 6k | 7k |
|---|---|---|---|---|---|---|
| baichuan2-7B | CMB-Exam | 29.38 | 30.64 | **31.84** | 31.50 | 31.54 |
| | MMCU-Medical | 27.67 | 28.52 | **29.37** | 28.77 | 29.01 |
| baichuan2-13B | CMB-Exam | 46.87 | 47.30 | **47.37** | 46.95 | 46.98 |
| | MMCU-Medical | 48.67 | 49.91 | **51.08** | 50.16 | 50.27 |
| Qwen1.5-7B | CMB-Exam | 60.47 | 60.45 | **61.96** | 60.78 | 60.53 |
| | MMCU-Medical | 63.64 | 63.92 | **66.09** | 64.49 | 64.10 |

Our results show that increasing the training data size initially boosts performance as the model learns to align with domain-specific knowledge. However, beyond a certain point (5K), performance degradation arise due to potential data redundancy and reduced diversity.

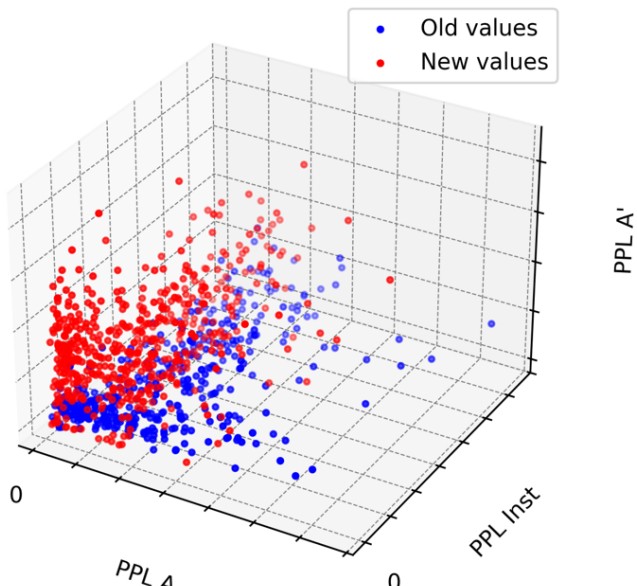

Figure 4: Domain shift before and after 3DS domain adaptation

## H  DOMAIN SHIFT ANALYSIS

To examine the domain shift effects induced by fine-tuning the model on the data subset selected via 3DS, we conduct an evaluation using a random sample of 500 examples from the entire domain dataset. Importantly, these examples are not necessarily included in the selected training dataset, allowing for an unbiased assessment of the model's domain adaptation. The decomposed difficulties of these samples are analyzed for the model before and after fine-tuning, as illustrated in Figure 4.

The figure reveals a clear shift in the point distribution towards reduced difficulty levels post fine-tuning. Specifically, the decrease in $PPL_\theta(Q)$ represents an improvement in the model's ability to comprehend instructions. Concurrently, the decrease in $PPL_\theta(A)$ indicates that the model has learned to generate more accurate answers. Interestingly, we also observe a slight increase in $PPL_\theta(A')$, which suggests the model exhibiting less confidence in its own responses. This could be interpreted as that the model becomes less overconfident after encountering new patterns in the domain-specific data. In addition, the more condensed distribution of points after domain adaptation indicates that the model has gained a more cohesive understanding of the domain, reducing the variance when handling domain samples.

Overall, these results demonstrate that the model has successfully adapted to the target domain, further validating the effectiveness of 3DS in facilitating domain adaptation.

## I  DOMAIN-SPECIFIC TASKS EVALUATION PROMPT

When evaluating model performance on the open Q&A dataset CMB-Clin, in addition to traditional metrics such as BLEU1, BLEU4 and Rouge, we conduct a pair-wise comparison to more thoroughly compare the fine-tuned models' medical analysis ability. In this experiment, we employ GPT-4, a highly capable LLM, as the judge to determine which model generates a better answer. Below, we present the prompt used in to instruct GPT-4 to compare the answers from two models in this qualitative pair-wise evaluation. To ensure a fair comparison and eliminate any possible positional bias in GPT-4, we randomly assign the answers from each model as "Student 1" or "Student 2" throughout the experiment.

> **GPT4 Evaluation Prompt**
>
> You are now a medical expert guiding students in analyzing medical cases. You have two students, Student 1 and Student 2. You assess them through real medical case questions and choose the one with the best answer to become your assistant.
>
> *[High-Quality Answer Criteria]*
> 1. The answer should address the question directly and solve the problem posed.
>
> 2. The description of symptoms should be comprehensive and accurate, and the diagnosis should be the most reasonable inference based on all relevant factors and possibilities.
>
> 3. The treatment recommendation should be effective and reliable, considering the severity or stage of the condition.
>
> 4. The prescription should consider indications, contraindications, and dosages, being both effective and reliable.
>
> *[Judgment Instructions]*
> Please compare the answers of Student 1 and Student 2. You need to tell me whether Student 1 is [better], [worse], or [equal] to Student 2. Compare their answers, refer to the question and the correct answer, and determine which one meets the given requirements more closely. Please only output one of the following: [Student 1 is better than Student 2], [Student 1 is worse than Student 2], or [Student 1 and Student 2 are equal]. Do not output any other words.
>
> *[Case Example]*
> Here is the [Question]:
> ```
> <Insert medical question here>
> ```
>
> Here is the [Standard Answer]:
> ```
> <Insert standard answer here>
> ```
>
> Here is [Student 1]'s answer:
> ```
> <Insert Student 1's answer here>
> ```
>
> Here is [Student 2]'s answer:
> ```
> <Insert Student 2's answer here>
> ```
> Please compare the two answers and give your judgment.

## J  BIAS AND FAIRNESS CONSIDERATIONS

Fairness and bias are critical considerations, particularly in sensitive domains like healthcare. While our approach demonstrates promising results in fine-tuning LLMs for medical tasks, it is essential to acknowledge its limitations and potential implications concerning fairness and bias. Our method employs the LLM to evaluate data quality and calculate data difficulty. Although the evaluation prompts and difficulty calculation metrics are designed to be neutral, the inherent biases in the base model may still influence the selection results. And the LoRA fine-tuning's impact on LLM fairness also needs further investigations Bui & Von Der Wense (2024). Another source of potential bias arises from the composition of our training data, which predominantly consists of Chinese medical texts. While this dataset effectively reflects the health conditions and medical practices of East Asian populations, it may limit the generalizability to other regions or demographics. Current LLM data selection methods generally prioritize factors such as difficulty, quality, or diversity, without addressing fairness or examine what data is included or excluded. They focus on improving model performance on standard benchmarks, while the impact of these methods on fairness, safety, and truthfulness benchmarks, such as SafetyBench (Zhang et al., 2024) and TruthfulQA (Lin et al., 2022), remains underexplored. Therefore, we recognize that these issues are valuable directions for

future research. Investigating how data selection and fine-tuning methods impact LLM fairness and safety will be essential for developing more equitable and reliable LLMs.

