# OpenReview forum: "3DS: Decomposed Difficulty Data Selection’s Case Study on LLM Medical Domain Adaptation"
_ICLR.cc/2025/Conference — ICLR 2025 Conference Withdrawn Submission_

### Official Review · Reviewer_rp16 · 2024-10-28

**Soundness:** 3
**Presentation:** 2
**Contribution:** 3
**Rating:** 5
**Confidence:** 3

**Summary:**

This work proposes an approach to selecting data likely to lead to effective adaptation of an LLM to a new domain. The approach proceeds in the two steps, with the first step using a prompt-based filter for data quality and the second step using a collection of entropy-based metrics that essentially assess the likelihood of the instruction, generated response and reference response under the model. They also offer an attention-weighting approach that weights tokens in the calculation based on their semantic importance, with the attention weights extracted directly from the attention layer of the transformer. Data are further filtered such that each of the entropy metrics fall into a pre-specified moderate range. A further contribution of the work is a new curated dataset of Chinese medical questions, to be released as a part of this work. In the experiments, they compare supervised tuning using all of this data with approaches that select subsets using the proposed approach and baselines. They find that the proposed approach improves performance.

**Strengths:**

* The approach does appear to be effective empirically, in that it leads to improvements in performance over baselines. Notably, they show that the method actually improves substantially over fine-tuning on all of the available fine-tuning data.
* The dataset contribution is significant and has potential for broad use.

**Weaknesses:**

* Regarding presentation, I generally found the description of the problem and the motivation for the proposed approach to be technically imprecise, with several of the core concepts not clearly defined or motivated. Overall, the work provides a reasonable heuristic and intuitive presentation of the problem and motivation that feels a bit “hand-wavy”, without clear theoretical grounding. This makes it difficult to address the soundness of the approach, regardless of whether it is empirically effective.
* The terminology used is technically imprecise. I have included some examples below:
  * It is not clear how a “model’s inherent knowledge distribution” or the “inherent preferences” are defined, nor what it means to align to it, nor why this might be relevant for identifying data for supervised fine-tuning.
  * Related to the above, the concept of “data difficulty” is not clearly defined. Furthermore, the work does not make a compelling case to motivate regarding why data difficulty is the key quantity to control for data selection. Ideally, there would be some clear technical justification for why a predefined middle range of the difficulty metrics is a good heuristic for selecting samples that would lead to effective supervised fine-tuning.
  * Some other examples of unclear terminology: “preference gaps” (lines 117-118), “learning requirements” (line 119), “target model” (line 161, 184, and elsewhere).
Misuse of terminology: The use of the term “domain adaptation” is not correct in this work. Domain adaptation has a specific technical meaning in machine learning that refers to adapting a classifier to a related domain without labeled data in that target domain (see Ben-David 2010 for example; https://dl.acm.org/doi/abs/10.1007/s10994-009-5152-4 ).

**Questions:**

* The work would be improved if the issues regarding the presentation raised in the weaknesses could be addressed.
* It would be helpful to elaborate on “k-center sampling based on instruction embeddings” (line 281-283).
* Threshold selection: “We determine the difficulty thresholds through experiments” (Line 355). Do you use a separate held-out validation set to select these thresholds, or are they selected on the test data? If they are selected on the test data, this would make me skeptical of the generalizability of the results.
* Given that the dataset contribution is a key contribution of the work, further details regarding the curation process and of the data itself would improve the work (in addition to what is currently included in Appendix B).
* RE: Instruction Understanding Difficulty: “A higher perplexity value indicates greater difficulty for the model to comprehend the instruction” (Line 221). Is this true? To my understanding, this is just a measure of the likelihood of the model of generating the instruction text. It is not clear whether this is actually a reasonable measure of whether the LLM comprehends the instruction.

---

> ### Author Response · Authors · 2024-11-22
>
> ## **W1: Regarding technical impreciseness**
> Thank you for your thoughtful feedback. Below, we address your concerns and provide our perspective on the theoretical context of LLM fine-tuning.
>
> We acknowledge the concern that some parts of our approach seem intuitive. However, we would like to point out that **the broader field of fine-tuning LLMs itself currently lacks a rigorous theoretical framework**. While fine-tuning is widely recognized as an essential step in building LLMs, its mechanisms remain underexplored, and the community is still actively investigating what types of data or methods are most effective. For instance, _Does Fine-Tuning LLMs on New Knowledge Encourage Hallucinations?_ [1] and _Learning or Self-Aligning? Rethinking Instruction Fine-Tuning_ [2] both empirically train models on different types of data and summarize the experiment results to deduce empirical conclusions, rather than provding riogorus theroretical analysis.
>
> Additionally, **in the narrower scope of data selection for fine-tuning LLMs, current methods are built based on intuitions and heurstics**. For instance, _DEITA_ [3] proposes a data selection framework purely based on heurstics and prompts; _From quantity to quality_ [4] also proposes a data difficulty metric of ppl(question|instruction)/ppl(question) based on intuitions. We argue that **our method utilizing perplexity to measure data difficulties has a degree of theoretical soundness rooted in the model uncertainty**.
>
> **Perplexity** is a well-defined and widely used metric that reflects a model's uncertainty in predicting token distributions. This metric is theoretically sound as it directly correlates with the entropy of the model's output probabilities. The decomposing of difficulties can find roots in human problem-solving models and metacognitive research, and the LLM's uncertainty is proved to be associated with its response correctness in uncertainty quantification works (_Semantic Uncertainty: Linguistic Invariances for Uncertainty Estimation in Natural Language Generation_ [5], _Shifting Attention to Relevance: Towards the Predictive Uncertainty Quantification of Free-Form Large Language Models_ [6], _MARS: Meaning-Aware Response Scoring for Uncertainty Estimation in Generative LLMs_ [7]).
>
> We agree that the lack of theoretical grounding in the broader fine-tuning domain creates challenges in assessing the soundness of approaches like ours. However, this challenge is not unique to our work, but is instead characteristic of an evolving field that heavily relies on empirical exploration. While further theoretical exploration is undoubtedly needed, we believe **our method provides a robust starting point for addressing the role and selection of fine-tuning data**.
>
> We appreciate your observation and concerns. However, we also emphasize that the fine-tuning of LLMs is currently an experimental field, as highlighted by recent literature. Our approach contributes to this area by proposing an interpretable and theoretically motivated method that has been empirically validated as effective.
>
> **(References are attached in Part 4)**

---

> > ### Author Response · Authors · 2024-11-22
> >
> > ## **W2: Terminology precision and clarity**
> > Thank you for your detailed feedback regarding terminology precision and clarity. Below, we address each of the issues you raised and provide additional justification and context.
> > ### **1. Model’s Inherent Knowledge Distribution and Inherent Preferences**
> > **Model’s Inherent Knowledge Distribution**
> >
> > Current research has shown that fine-tuning LLMs on data that exceeds their existing knowledge or includes concepts entirely unfamiliar to them can lead to suboptimal performance. For example:  _Does Fine-Tuning LLMs on New Knowledge Encourage Hallucinations?_ [1] and _Learning or Self-Aligning?_ [2] demonstrate that fine-tuning on data beyond the LLM's knowldege acquired during pretraining leads to serious hallucination and performance degredation. However, fine-tuning on fully known data also yields suboptimal results. Therefore, **model’s inherent knowledge distribution in our paper refers to the knowledge the model has already internalized in its parameters** (thus "inherent") during pretraining. In our work, we use perplexity to measure the alignment of data with this inherent knowledge distribution, identifying moderately challenging data for fine-tuning.
> >
> > **Inherent Preference**
> >
> > The concept of preferences comes from **preference optimization** (DPO [8], RLHF [9]), which refers to the judgement by either human or models to decide what data is deemed good data. In _Self-Rewarding Language Models_ [10], the LLM to be trained is prompted to score data and provide reward feedback for its alignment. **The scoring process and the data scores reflect the evaluator's preferences for data**. In this context, the prompting and scoring mechanism in our Stage 1 reflects the model's inherent preferences for data. Utilizing the high-scored data for fine-tuning is aligning the model with its self preference.
> > ### **2. Data Difficulty**
> > **The Definition of Data Difficulty**
> >
> > Data difficulty is a widely used concept in data selection and data pruning. _Beyond Neural Scaling Laws_ [11] defines difficult data as data points that deviates from cluster centroids and emphasizes the importance of learning on these data to improve model generalizability. _From Quantity to Quality_ [4] and _Superfiltering_ [12] proposes Inruction-following difficulty and select most difficult data to fine-tune the LLM. Data complexity is also proposed as an important selection metric in _DeITA_ [3].
> >
> > It is true that there lacks a universal definition, yet data difficulty is still a widely addressed metric. In our work, **data difficulty refers to the degree to which the model has mastered the data**, calculated using perplexity.
> >
> > **Why Difficulty is Key and  Why Moderate Difficulty**
> >
> > As mentioned in the response regarding Inherent Knowledge Distribution, **previous studies show that the model can not learn well on fully known and completely unknown data**. Therefore, identifying to what extent the model knows the data is key to data selection, and this knowledge degree is defined as data difficulty in our approach. Additionally, anthoer previous study _When Less is More_ tests several selection metrics for LLM pre-training and discovers that perplexity-based selection produces the most effective results, further strengths our insights of selection based on difficulty calculated via perplexity. Moreover, it finds that focusing on data with middle perplexity improves training outcomes most significantly. Our middle range selection is built on these prior insights and is empirically proven to be effective in our experiments. Finally, as mentioned in the response to Weakness 1, the internal mechanism of LLM fine-tuning is still under exploration and clear theoretical justification needs further studies.
> > ### **3. Clarifications on unclear terminologies**
> > - **Preference Gaps**:The term refers to the misalignment between the scoring preferences of an external model (e.g., GPT-4 or a reward model) and the preferences of the target model. For example, GPT-4 may rate certain data as high quality based on its understanding and criteria, but these preferences may not align with preferences of the model to be trained.
> > - **Learning Requirements**:We acknowledge that this terminology could have been clearer and apologize for it. What we mean is that for optimal performance, a model needs to train on data that is neither too easy nor too complex. It is the requirement for beneficial moderately difficult data during model fine-tuning.
> > - **Target Model**: This term refers to the model to be fine-tuned on the selected data in our experiments. We have revised the paper to ensure consistency and clarify this definition.
> >
> > **Due to the length constraints, the remaining part regarding "Domain Adaptation" is attached in the following comment**
> >
> > **(References are attached in Part 4)**

---

> > > ### Author Response · Authors · 2024-11-22
> > >
> > > ## **W2: Terminology precision and clarity (cont.)**
> > > ### **4. Use of “Domain Adaptation”**
> > > While traditionally, domain adaptation indeed refers to adapting a classifier to a related domain without labeled data in the target domain, the term has evolved. **In the context of LLMs, it can be used to describe training an LLM to enhance its domain knowledge and abilities**, which is widely used in current research. For example:
> > > - _Adapting Large Language Models via Reading Comprehension_ [13] discusses **domain-adaptive pretraining** to specialize LLMs for specific domains like biomedicine, finance and law.
> > > - _Huatuo GPT-II_ [14] explicitly states: “**Adapting a language model into a specific domain, a.k.a. ‘domain adaptation**,’ is a common practice when specialized knowledge, e.g., medicine, is not encapsulated in a general language model.”  and fine-tunes the LLM in domain-specific instruction tuning data.
> > > - NVIDIA’s _ChipNeMo_ [15] also includes **"domain adaptive continued pretraining, model alignment with domain-specific instructions" into domain adaptation methods**.
> > >
> > > In the context of LLMs, domain adaptation has come to describe the process of training general-purpose LLMs to specialize in specific domains. Our use of the term follows this emerging convention.
> > >
> > > We hope our clarification has addressed your concerns.
> > >
> > > ## **Q1: Regarding presentation**
> > > We have addresses the issues in the above responses to Weakness.
> > >
> > > ## **Q2: K-Center sampling**
> > > Thank you for your kind suggestion. We provide the following clarification.
> > > The process is applied through:
> > > 1. **Instruction Embedding Generation**: The instruction part of each data sample is encoded into an embedding using the LLM to represent its semantics. Specifically, we extract the last hidden states of the LLM and compute the average across all tokens in the sequence to generate a fixed-size embedding vector.
> > > 2. **K-Center Sampling Algorithm**: Using the generated embeddings, the k-center sampling algorithm greedily selects \( k \) data points that are as distinct as possible by maximizing the minimum distance between any pair of selected embeddings.
> > >
> > > This detailed introduction, along with pseudo-codes, have been added to the **Appendix C** for better readability and clarification.
> > >
> > > ## **Q3: Threshold selection**
> > > The difficulty thresholds were selected using a separate CMB-validation set containing 280 samples provided by the CMB benchmark. We conducted experiments across various threshold ranges to identify the optimal configuration for our approach and selected the best-performing thresholds. The results of these experiments on the validation set are summarized below:
> > >
> > > | **Model**       | **5-55** | **10-60** | **15-65** | **20-70** | **25-75** | **30-80** |
> > > |------------------|----------|-----------|-----------|-----------|-----------|-----------|
> > > | Baichuan2-7B     | 28.22    | **28.99** | 27.67     | 28.22     | 24.18     | 24.25     |
> > > | Baichuan2-13B    | 37.58    | 36.83     | **37.81** | 37.63     | 38.58     | 37.65     |
> > > | Qwen1.5-7B       | 56.97    | 57.60     | 57.80     | 58.23     | **58.65** | 57.91     |
> > >
> > > To further address concerns about generalizability, we note that our training and test sets are entirely out-of-domain (OOD). The models are trained on a diverse medical instruction-tuning dataset and evaluated on two diiferent medical multiple-choice test sets MMCU-Medical and CMB-Exam, and an open-ended medical analysis task CMB-clin.
> > >
> > > Our consistent results on these OOD test sets further demonstrate the robustness and generalizability of our approach.
> > >
> > > We hope these explanations help clarify our results and we have added these settings to **Appendix E** in our revised paper. If you have additional questions or suggestions, we are happy to address them.
> > >
> > > ## **Q4: Regarding the dataset**
> > > Thank you for your valuable feedback. In response, we have added further details to **Appendix B** of the revised paper. Specifically, we have included:
> > >
> > >   1. **A detailed description of the dataset curation process**. We have included information on how the data was collected and processed.
> > >   2. **Representative data samples**: We have provided concrete examples of the data to give readers a better understanding of its structure and content.
> > >   3. **A Datasheet for the Dataset**: Following practices in Datasheets for dataset, we have included a datasheet that summarizes key details about the dataset, such as its intended use, composition, and any potential limitations.
> > >
> > > Additionally, our dataset has been made publicly available at [https://drive.google.com/drive/folders/1SfrwQkDrQJ8i_EIqfc2Di0Xa5Y5pzY9H ], ensuring transparency and accessibility for the research community.
> > >
> > > **(References are attached in Part 4)**

---

> > > > ### Author Response · Authors · 2024-11-22
> > > >
> > > > ## **Q5: RE: Instruction Understanding Difficulty**
> > > > Thank you for your inquiry about the utility of perplexity as the instruction understanding difficulty metric. You are correct that perplexity measures how well a model can predict a sequence of words, thus reflecting its textual predictability. Nonetheless, **recent research has begun to show that perplexity also indicates the model’s familiarity with the content of prompts, and consequently, its understanding**.
> > > >
> > > > * The study _Demystifying Prompts in Language Models via Perplexity Estimation_ [16] posits that the effectiveness of a prompt is closely linked to the model’s familiarity with the language used in the prompt. It empirically shows that prompts with lower perplexity scores tend to yield better performance. This suggests that **a lower perplexity score indicates a higher degree of familiarity or comprehension by the model**, supporting the idea that a model with lower perplexity in response to an instruction likely has a better grasp of the requested task, thereby enhancing task execution. (As referenced in lines 211-213 of our paper.)
> > > >
> > > > * Building on this concept, the research presented in _Revisiting Demonstration Selection Strategies in In-Context Learning_ [17] examines how the choice of demonstrations influences performance in in-context learning scenarios. The findings show that demonstrations leading to **lower perplexity** scores, when used as part of instructional context, **enhance the model’s task performance**. This indicates that lower perplexity is associated with a more profound understanding of the task requirements by the model.
> > > >
> > > > These studies collectively suggest that within the framework of prompting and following instructions in large language models, **a lower perplexity score can indeed serve as a reasonable proxy for the model’s comprehension of the instructions**. This is attributed to the fact that lower perplexity indicates the instructional language lies well within the model’s pre-existing knowledge base, which implies a more robust grasp of the task at hand.
> > > >
> > > > Additionally, in _Superfiltering_ [12], the authors note that perplexity comes from the model's exposure to the underlying distribution of language during pretraining, so **this likelihood can interpret the model's familiarity with the given text**. In _Adapting Large Language Models via Reading Comprehension_ [13], the token likelihood of “predict-the-next-token/sentence” manner is used to probe the model's domain knowledge. Similarly, **perplexity built on token likelihood can also reveal how well the model understands the knowledge in the given text**. Given the above, we argue that perplexity serves as a reasonable and interpretable measure of instruction difficulty. While it does not explicitly model "comprehension" as a human would, it effectively captures the model's probabilistic uncertainty and familiarity with the instruction text.
> > > >
> > > > **In conclusion**, recent studies support the view that perplexity, beyond simply measuring the likelihood of language generation, can reflect the model's familiarity with instructional content. This makes it a valuable indicator of how well a model understands and is likely to perform well on a given task. This perspective validates the claim made in our work that higher perplexity values suggest greater difficulty for the model in comprehending instructions.
> > > >
> > > > ## **References**
> > > >
> > > > [1] Gekhman, Zorik, et al. Does Fine-Tuning LLMs on New Knowledge Encourage Hallucinations?. EMNLP (2024).
> > > >
> > > > [2] Ren, Mengjie, et al. Learning or self-aligning? rethinking instruction fine-tuning. ACL (2024).
> > > >
> > > > [3] Liu, Wei, et al. What makes good data for alignment? a comprehensive study of automatic data selection in instruction tuning." ICLR (2024).
> > > >
> > > > [4] Li, Ming, et al. From quantity to quality: Boosting llm performance with self-guided data selection for instruction tuning. NAACL (2024).
> > > >
> > > > [5] Kuhn, Lorenz, Yarin Gal, and Sebastian Farquhar. Semantic uncertainty: Linguistic invariances for uncertainty estimation in natural language generation. ICLR (2023).
> > > >
> > > > [6] Duan, Jinhao, et al. Shifting attention to relevance: Towards the predictive uncertainty quantification of free-form large language models. ACL (2024).
> > > >
> > > > [7] Bakman, Yavuz Faruk, et al. MARS: Meaning-Aware Response Scoring for Uncertainty Estimation in Generative LLMs. ACL (2024).
> > > >
> > > > [8] Rafailov, Rafael, et al. Direct preference optimization: Your language model is secretly a reward model. NIPS (2024).
> > > >
> > > > [9] Achiam, Josh, et al. Gpt-4 technical report. arXiv (2023).
> > > >
> > > > [10] Yuan, Weizhe, et al. Self-rewarding language models. ICML (2024).
> > > >
> > > > [11] Sorscher, Ben, et al. Beyond neural scaling laws: beating power law scaling via data pruning. NIPS (2022).
> > > >
> > > > [12] Li, Ming, et al. Superfiltering: Weak-to-strong data filtering for fast instruction-tuning. ACL (2024).
> > > >
> > > > [13] Cheng, Daixuan, Shaohan Huang, and Furu Wei. Adapting large language models via reading comprehension. ICLR (2024).

---

> > > > > ### Author Response · Authors · 2024-11-22
> > > > >
> > > > > [14] Chen, Junying, et al. Huatuogpt-ii, one-stage training for medical adaption of llms. COLM (2024).
> > > > >
> > > > > [15] Liu, Mingjie, et al. Chipnemo: Domain-adapted llms for chip design. arXiv (2023).
> > > > >
> > > > > [16] Gonen, Hila, et al. Demystifying prompts in language models via perplexity estimation. EMNLP (2022).
> > > > >
> > > > > [17] Peng, Keqin, et al. Revisiting demonstration selection strategies in in-context learning." ACL (2024).

---

> > > > > > ### Comment · Reviewer_rp16 · 2024-11-25
> > > > > > **Response to rebuttal**
> > > > > >
> > > > > > Thank you for the detailed rebuttal to the concerns raised in my review. On the whole, the clarifications are helpful and improve my understanding and reception of the paper. I have read the updated manuscript and appreciate the cases where additional context and definitions were added. However, I still believe some additional revision to bring technical precision to key concepts would greatly improve the paper. I have updated my overall score (3->5), Soundness score (2->3), and Presentation score (1->2). I have included responses to individual points below:
> > > > > >
> > > > > > * W1: I agree with the authors that it is not an issue for this work that the field is empirical and lacks rigorous theoretical foundation. My concerns are not so much about the empirical nature of the field and this study, but rather the need to precisely describe and ground the motivation, research question, hypotheses, and quantities measured.
> > > > > > * W2: The lack of technical imprecision with regards to key terminology was a key issue for me. The authors' revisions regarding the model’s inherent knowledge distribution and preferences, as well as the comments about data difficulty, are helpful. If accepted, I would encourage the authors to continue to revise the paper for the camera-ready submission to include more of the context that they present in the rebuttal in the paper itself.
> > > > > >   * Regarding the point on domain adaptation, I concede that it is not entirely unreasonable to call this domain adaptation given its usage elsewhere in the literature. I would argue that something like “domain adaptive fine-tuning” would be better, however.
> > > > > > * Q2-4: Thank you for adding the details regarding K-center sampling, threshold selection, and the dataset.
> > > > > > * Q5: Thank you for the context on the prior work regarding the use of perplexity for quantifying instruction understanding. I agree with the author’s that this interpretation is reasonable given the prior work.

---

> > > > > > > ### Author Response · Authors · 2024-11-27
> > > > > > >
> > > > > > > Dear Reviewer rp16,
> > > > > > >
> > > > > > > We would like to express our sincere gratitude to you for the time and effort taken to re-evaluate our work and appreciate your valuable suggestions provided.
> > > > > > > Below is a point-by-point response to your comments:
> > > > > > >
> > > > > > > #### **W1: Motivation, Research Question, Hypotheses, and Quantities Measured**
> > > > > > > We agree with you that the research question, hypotheses, and quantities measured in the paper could be described and grounded more precisely. We will make sure to revise the paper in the camera-ready version if accepted, ensuring that each of these elements is clearly grounded in a more detailed theoretical context.
> > > > > > >
> > > > > > > #### **W2: Terminology Precision**
> > > > > > > We understand the importance of providing clear and precise definitions of critical concepts. In the newly revised paper, we have added explanations to the model's inherent knowledge distribution, preferences, and data difficulty, in order to clarify these concepts. We will further improve the paper by choosing more precise terminologies and adding additional theoretical grounding to ensure both precision and better reader comprehension in the camera-ready version. We will also take extra care to ensure that all technical terminology is used consistently and accurately.
> > > > > > >
> > > > > > > As for the use of "domain adaptation", we agree that the term "domain adaptive fine-tuning" is more precise and less ambiguous. Given that the term "domain adaptation" has already been used in the title of the paper, we will update the terminology throughout the paper to "domain adaptive fine-tuning" in the camera-ready version to ensure consistency.
> > > > > > >
> > > > > > > We would like to thank you once more for your acknowledgement and constructive suggestions.
> > > > > > >
> > > > > > > Sincerely,
> > > > > > >
> > > > > > > Authors of Paper 10359

---

> > > > > > > > ### Author Response · Authors · 2024-12-03
> > > > > > > >
> > > > > > > > Dear Reviewer rp16,
> > > > > > > >
> > > > > > > > As the rebuttal period is quickly drawing to an end, we would like to take this opportunity to thank you for your thorough review of our paper, as well as for your thoughtful questions and suggestions.
> > > > > > > >
> > > > > > > > If you have any further feedback or insights, we would be most appreciative if you could share them with us. We are also happy to provide additional clarification on any remaining points or issues.
> > > > > > > >
> > > > > > > > Once again, thank you for your time, attention, and valuable suggestions to our work!
> > > > > > > >
> > > > > > > > Best regards,
> > > > > > > >
> > > > > > > > Authors of Paper 10359

---

### Official Review · Reviewer_oddX · 2024-11-03

**Soundness:** 3
**Presentation:** 3
**Contribution:** 3
**Rating:** 6
**Confidence:** 3

**Summary:**

The paper proposes a model-centric approach for data selection, which can be used for supervised fine tuning of LLMs. The authors identify a gap in the existing data selection strategies, which can lead to suboptimal model performance and propose 3DS, a strategy that uses the model’s inherent knowledge during data selection, while also accounting for 3 kinds of data difficulty – related to instruction understanding, response confidence and correctness. The paper identifies a way to tackle a known problem, related to data selection in supervised fine tuning of LLMs.

**Strengths:**

•	Significance: The authors have proposed a solution for a known problem in supervised fine tuning, specific to the medical domain.

•	Clarity: The paper is quite well written and relatively easy to follow along. Clarifications needed in the draft are mentioned in the Questions section of the review.

•	Quality: The authors have evaluated their framework on 3 different datasets.

•	Originality: The use of attention based weighting is an interesting approach to tackle the relative semantic importance of tokens.

•	Others: The limitations have also been acknowledged.

**Weaknesses:**

•	The authors have provided a link for the code and data, but it is not yet available publicly. Making the data available publicly would be greatly beneficial to the community.

•	Since the authors propose the 3DS data selection framework as an alternate to GPT-4 annotation based or manual data selection based strategies, it would be good to include a comparison with these techniques in the experiments.

•	Since the focus is on medical domain and LLMs, there needs to be a comparison against LLMs developed / tuned for the medical domain, like MedPALM[1], MediTron[2] etc.

[1] Singhal K, Tu T, Gottweis J, Sayres R, Wulczyn E, Hou L, Clark K, Pfohl S, Cole-Lewis H, Neal D, Schaekermann M. Towards expert-level medical question answering with large language models. arXiv preprint arXiv:2305.09617. 2023 May 16.

[2] Chen Z, Cano AH, Romanou A, Bonnet A, Matoba K, Salvi F, Pagliardini M, Fan S, Köpf A, Mohtashami A, Sallinen A. Meditron-70b: Scaling medical pretraining for large language models. arXiv preprint arXiv:2311.16079. 2023 Nov 27.

**Questions:**

•	The authors mention “the model’s inherent knowledge distribution” in the introduction (Line 056, 057) – it would be good to clarify if they are referring to distribution with respect to labels, or tasks being considered.

•	How do the authors ensure that the data selected by 3DS indeed follows the “model’s inherent knowledge distribution”?

•	What is the intuition behind having less diverse data in the data selected (this is mentioned as an issue of prior data selection strategies)? (Line 055) Does the addition of data similar to the inherent knowledge distribution end up reinforcing what the model already knows?

•	In Stage I, the target model is used to provide a data quality score (detailed in Appendix A). However it is a bit unclear if the overall score is a combination of the 5 dimensions listed in Appendix A. Please provide a reference for why the 5 dimensions listed in the Appendix are good indicators of data quality.

•	Lines 198-206: The authors identify 3 components from general problem solving – it would be good to add a reference/citation for why these 3 aspects are crucial and the basis of consideration in this paper.

•	In Equation 5, how is the aggregate function defined?

•	Lines 278-283: how does the selection ensure non-redundant data (already known to the model) is not captured? Please include details of k-centre sampling for better readability.

•	In Algorithm 1, were there any situations when $S_{mid}$ was an empty set? Was this also a consideration in choosing $\tau^{low}$ and $\tau^{high}$?

•	It would be good to have an experiment showing sensitivity to training data budget (currently set to 5K, line 351).

•	In Table 3, kindly clarify the metric used for comparison.

•	Please include details (maybe in the Appendix) on selection of hyperparameters, steps taken to ensure reproducibility of results.

---

> ### Author Response · Authors · 2024-11-22
>
> ## **W1: Availability of codes and data**
> We are pleased to inform you that the codes and data have now been publicly accessible.
>   - The codes are at: [https://anonymous.4open.science/r/3DS-E67F]
>   - The data is at: [https://drive.google.com/drive/folders/1SfrwQkDrQJ8i_EIqfc2Di0Xa5Y5pzY9H]
>
> We hope this will be beneficial to the community and facilitate further research.
> ## **W2: Comparison with proprietary LLM annotation**
> Thank you for your suggestion.
>
> Our proposed automated selection framework aims to reduce the cost and effort associated with data annotation and selection. Therefore, we did not initially include comparisons with costly GPT-4 or manual annotation. However, in response to your comment, we conducted additional experiments using powerful external LLMs and provide following analysis.
>
> We first used **GPT-4o** to evaluate data quality using the same prompt in our paper. We annotated 1,000 samples at a cost of 14 dollars, among which only 100 received a score $\geq$ 85. To obtain 5,000 high-quality samples (matching the data budget in our paper), the estimated cost would be at least 700 dollars, which exceeds our budgetary constraints.
>
> Therefore, we provide **a cost-effective alternative, using Qwen2.5-72B**, the latest, and most powerful open-sourced LLM in Qwen series as a substitute for GPT-4o. Using Qwen2.5-72B, we annotated the training dataset and selected 5,000 data samples with scores &\geq& 85 to train models. We compared these models against those trained using our 3DS framework. The results are presented below:
>
> | **Dataset**          | **CMB-Exam**       |                     | **MMCU-Medical**    |                     |
> |------------------|--------------------|---------------------|---------------------|---------------------|
> | **Model**            | QwenRate           | 3DS                 | QwenRate            | 3DS                 |
> | Baichuan2-7B     | 28.55              | **31.84**               | 25.90               | **29.37**               |
> | Baichuan2-13B    | 46.82              | **47.37**               | 50.55               | **51.08**               |
> | Qwen1.5-7B       | 60.67              | **61.96**               | 64.17               | **66.09**               |
>
>
> Across three models and two datasets, the models trained with data selected using 3DS consistently outperformed those trained with data annotated by Qwen2.5-72B. These results demonstrate the superiority of our model-centric data selection approach, which identifies suitable data based on the model itself without incurring additional annotation costs.
>
> We have included this analysis in the **Appendix G.2** of our revised paper to provide a detailed comparison and to highlight the necessity of model-centric data selection. Thank you again for your insightful feedback.
>
> ## **W3: Comparison with existing medical LLMs**
> Thank you for pointing out the importance of comparing our approach against existing medical LLMs. Below, we present the results of our additional experiments.
>
> We found that **Med-PaLM2 is currently available only to a limited number of Google Cloud customers** and unfortunately could not access it for evaluation. However, we compared our approach against the **open-source MediTron** (7B version due to its similar size to Baichuan2-7B and Qwen1.5-7B) and **two state-of-the-art Chinese medical LLMs, HuatuoGPT-II-7B and 34B** (HuatuoGPT-II [1]).
> | **Model**             | **CMB-Exam** | **MMCU-Medical** |
> |------------------------|--------------|------------------|
> | Baichuan2-7B-3DS      | 31.84        | 29.37           |
> | Baichuan2-13B-3DS     | 47.37        | 51.08           |
> | Qwen1.5-7B-3DS        | **61.96**    | _66.09_         |
> | Meditron-7B           | 11.20        | 12.16           |
> | Huatuo-7B             | 27.69        | 47.18           |
> | Huatuo-34B            | _59.54_      | **66.10**       |
>
> The results are summarized below:
> * **MediTron-7B**, being an English LLM, performs poorly on Chinese medical benchmarks, significantly underperforming our method.
> * **Huatuo-7B** shows strong performance on MMCU, outperforming Baichuan2-7B fine-tuned with our 3DS. However, it performs poorly on the more complex and larger CMB-Exam dataset, where it is surpassed by all models fine-tuned using 3DS.
> * **Huatuo-34B**, with nearly five times the model size of Qwen1.5-7B, achieves comparable performance, demonstrating the competitiveness of our 3DS.
>
> It's worth noting that the performance of fine-tuned models is closely tied to its original capability, so the **relative improvements achieved through domain-specific fine-tuning are more important than absolute performance**. Still, the superior results achieved by 3DS demonstrate that it's a practical and efficient way to develop medical LLMs.
>
> We have included these results and analysis in **Appendix G.3** of our revised paper. Thank you again for raising this important point.
>
> **(References are attached in Part 5)**

---

> > ### Author Response · Authors · 2024-11-22
> >
> > ## **Q1: Model’s inherent knowledge distribution**
> > We refer to this distribution as the broad, task-agnostic factual knowledge embedded in the model’s parameters during pre-training on large, diverse textual corpora.
> >
> > This knowledge is not organized or distributed according to specific tasks or labels. Instead, it is embedded as a generalized knowledge base within the model’s parameters, without being explicitly tied to any particular downstream tasks. Rather, the model draws from this broad knowledge base as needed during inference.
> >
> > To support this perspective, we cite the following works:
> > * _Language models as knowledge bases?_ [2]
> > * _A review on language models as knowledge bases._ [3]
> > * _Crawling the internal knowledgebase of language models._ [4]
> >
> > In conclusion, by "inherent knowledge distribution" we refer to broad factual information embedded in the LLM’s parameters, not specific to any tasks or labels. Our work focuses on adapting this knowledge base to specific domain to enhance its domain performance.
> >
> > ## **Q2: How to ensure data follows the “model’s inherent knowledge distribution”?**
> > The core idea is **selection based on the knowledge distribution of the current model itself**.
> >
> > In **Stage 1**, we prompt the model to select high-quality data based on its knowledge (what it knows) and preference (what it thinks is good). Previous research _Self-Distillation Bridges Distribution Gap in Language Model Fine-Tuning_ [5] finds that prompting the model to rewrite data based on its knowledge effectively aligns data with its distribution and mitigates catastrophic forgetting and improves performance during fine-tuning.  Additionally, _Self-Rewarding Language Models_ [6] demonstrates that the model itself can be prompted to assess data quality and the resulted data is well-aligned with its own preference and yields satisfying results in preference optimization. These findings support our strategy of prompting the model to select data based on its knowledge and preference. By doing so, we ensure that the chosen data is aligned with its knowledge distribution.
> >
> > In **Stage 2**, we further select data using **perpleixty-base difficulties**. In _Adapting Large Language Models via Reading Comprehension_ [7], the token likelyhood is used to probe the model's domain knowledge. Similarly, perplexity built on token likelihood can also reveal model's knowledge distribution. In _From Quantity to Quality_ [8], Instruction-Following Difficulty is also designed based on perplexity. These findings support that **our difficulty metrics built on perplexity can serve as an indicator of how misaligned the data is with the model's knowledge, ensuring that the selection is based on the model's knowledge disrtibution.**
> >
> > ## **Q3: Regarding diversity**
> > Thank you for your question. To clarify, our concern here is not that diversity itself is problematic, but rather that prior strategies often **define diversity and high-quality data without considering the model's current knowledge**. They overly emphasize broad coverage across scenarios, which may lead to the inclusion of data that may hinder model learning.
> >
> > The primary goal of data diversity in fine-tuning is to ensure that the model is exposed to a wide range of relevant scenarios, avoiding overfitting to a narrow set of data. However, diversity should be meaningful, not just broad. Specifically, domain adaptation fine-tuning requires more targeted alignment with the model’s pre-existing knowledge rather than introducing entirely new or drastically different examples.
> >
> > Previous studies _Does Fine-Tuning LLMs on New Knowledge Encourage Hallucinations?_ [9] and _Learning or Self-Aligning?_ [10] discover that fine-tuning on data beyond the model's knowledge leads to significant performance degradation, and fine-tuning on data fully mastered by the model also yields suboptimal results. They both highlights the importance of **appropriately aligning the model with data similar to its inherent knowledge distribution (not identical)**. In the supervised fine-tuning phase, the goal is not to teach the model entirely new knowledge, but rather to ensure that the model becomes better aligned with the specific domain without forgetting. Thus, our method of introducing data that is **similar (not identical as we select middle difficulty data)**  to the model’s internal knowledge — rather than depending solely on diversity — is a better strategy for effective domain adaptation.
> >
> > Additionally,  our framework does not fully discard diversity. We incorporate **diversity selection k-center sampling** to ensure that while the selected data is aligned with the model’s knowledge, it is not overly similar and redundant. Our approach selects data that is both aligned with the model’s knowledge distribution and sufficiently diverse to prevent overfitting, striking the right balance between effective adaptation and generalization.
> >
> > **(References are attached in Part 5)**

---

> > > ### Author Response · Authors · 2024-11-22
> > >
> > > ## **Q4: Data quality evaluation and dimensions**
> > > The data quality score is determined by the model itself. It is prompted to evaluate data along the 5 dimensions and directly provide an overall score without explicitly outputting intermediate judgments.
> > >
> > > Below, we provide a detailed explanation of how these dimensions are aligned with recent findings and why they are reasonable indicators of data quality.
> > >   1. **Question Complexity**: The importance of **instruction complexity** as a key metric for data selection has been established in _DeITA: What Makes Good Data for Alignment?_ [11]. The paper emphasizes that selecting data with high instruction complexity leads to more efficient fine-tuning. Following this insight, we include quesition complexity as one dimension.
> > >   2. **Response Relevance**: _CMB_ [12], which proposes a medical benchmark including CMB-clin (an open-ended medical QA task), highlights relevance as a key metric when using GPT-4 to evaluate LLM responses. And _HuatuoGPT-ii_ [1] also emphasizes that a response should be orientated towards the question during evaluation. Thus, we include response relevance to ensure the model's outputs are relevant to the given instruction.
> > >   3. **Response Thoroughness**: Thoroughness reflects the completeness of the model's response. Again, _CMB_ includes completeness as a key metric during evaluation. Response thoroughness is critical for determining the quality of a response, particularly in domains like healthcare where incomplete responses can have significant consequences.
> > >   4. **Response Logic**: As stated in _Huatuo GPT-II_, logical clarity is one of the central metrics for evaluation. This metric ensures that responses are coherent and logically consistent. Therefore, we also include response logic as a core dimension.
> > >   5. **Knowledge Richness**: In the medical domain, the accuracy, professionalism, and depth of knowledge in responses are crucial. _Huatuo GPT-II_ evaluates responses according to its "richness" and "professionalism and accuracy". _CMB_ also emphasizes "medical proficiency". Following these insights, we include knowledge richness to capture the knowledge depth and accuracy of the content provided in the response.
> > >
> > > By integrating the above insights, our 5 data quality dimensions—**question complexity, response relevance, response thoroughness, response logic, and knowledge richness**—are well-grounded in recent research. We hope this explanation clarifies the rationales behind our data quality prompting method.
> > >
> > > ## **Q5: Rationales for 3 decomposed difficulty components**
> > > Thank you for your insightful question.
> > >
> > > Widely recognized frameworks for problem-solving, such as Pólya’s _How to Solve It_ [13] and _PISA 2012 Results: Creative Problem Solving_ [14], decompose problem-solving into 4 stages. Polya's framework consists of understanding the problem, devising a plan, Carrying out the plan, looking back to reflect on the solution process. PISA 2012 framework consists of exploring and understanding, representing and formulating, planning and executing, monitoring and reflecting. Both emphasize that problem-solving **starts with understanding the problem, followed by planning, executing, and evaluating the solution**.
> > >
> > > Our proposed three aspects introduced in lines 198-206 are closely aligned with these foundational insights. The "**understanding the problem"** component and deduced **Instruction Understanding Difficulty** correspond directly to the first step in both Pólya’s and PISA’s frameworks. The subsequent components "assessing confidence in the solution, and finally providing the answer" and corresponding **Response Confidence Difficulty and Response Correctness Difficulty** can be refered to as the 2-4 steps described in Pólya’s and PISA’s frameworks, as we allow the model to generate its own response and evaluate its confidence and correctness, similar to the **planning, executing, and evaluating process**.
> > >
> > > The evaluation on both response confidence and correctness is further supported by findings in _Assessing Confidence and Certainty of Students in an Undergraduate Linear Algebra Course_ [15]. This study highlights that correctness alone is insufficient to assess a learner's proficiency, as it may include lucky answers. And evaluating the learner's confidence helps provide a more comprehensive understanding of the learner's knowledge mastery. In our framework, Response Confidence Difficulty reflecting the model's confidence in the response and Response Correctness Difficulty reflecting the model's ability to provide correct responses serve analogous purposes.
> > >
> > > In summary, our proposed three components are grounded in established human problem-solving models and supported by findings in metacognitive research. We have also added corresponding references to our revised paper to strengthen the rationale for our approach. Thank you again for your insightful feedback.
> > >
> > > **(References are attached in Part 5)**

---

> > > > ### Author Response · Authors · 2024-11-22
> > > >
> > > > ## **Q6: Aggregate function in Equation 5**
> > > > The aggregate function is defined as either the **mean** or **max** of the attention scores:
> > > >   - **Mean**: The importance of token $t_i$ is computed as the average of the attention scores from all subsequent tokens $t_j (j > i)$.
> > > >   - **Max**: The importance of token $t_i$ is computed as the maximum attention score from any subsequent token $t_j (j > i)$.
> > > >
> > > > We hope this explanation clarifies the implementation.
> > > >
> > > > ## **Q7: How to ensure non-redundant data**
> > > > Redundancy in data already known to the model is addressed by filtering based on decomposed difficulties. We argue that the designed perplexity-based difficulties reflect the model's understanding of the data. **By removing low-difficulty data with metrics below the lower thresholds, data already mastered by the model is excluded**. While this filtering may not be perfect and could occasionally include some easy data, our empirical experiments indicate that focusing on data with moderate difficulty yields better performance. The experiments in Section 5 (lines 459-475) show that lowering the difficulty bounds to include easier data leads to performance degradation. Therefore, we argue that our approach effectively mitigates this redundancy and prioritizes data that is most beneficial for fine-tuning.
> > > >
> > > > The **K-Center sampling algorithm** selects a diverse subset of data by maximizing the minimum distance between selected samples in the embedding space. The instruction part of each data sample is encoded into an embedding through the LLM. Specifically, the last hidden states of the LLM are averaged across tokens to serve as the input features for K-Center. Using these embeddings, the algorithm then selects $k$ data points that are as distinct from each other as possible by greedily maximizing the minimum distance between any selected pair.
> > > >
> > > > The detailed introduction to this process, along with pseudo-codes have been added to **Appendix C** of the revised paper for better readability. Thank you for your kind reminder.
> > > >
> > > > ## **Q8: Regarding $S_{mid}$**
> > > > It is indeed possible for $S_{mid}$ to be an empty set if the difficulty lower bound $tau_{low}$ and upper bound $tau_{high}$ are set too close to each other, such as 53%-55%. Although $tau_{low}$ and $tau_{high}$ are determined as percentiles of the difficulty distribution, ensuring that for any single difficulty metric there is always data within the specified middle range, the intersection of  middle ranges from the three difficulty metrics can potentially yield an empty set.
> > > >
> > > > In our main experiments, we set the middle range to 50%, which ensures a sufficiently broad overlap across the metrics. With this configuration, the filtered set $S_{mid}$ typically contains 8,000 to 9,000 samples.
> > > >
> > > > ## **Q9: Ablation on data budget**
> > > > Thank you for your suggestion regarding the sensitivity of 3DS to the training data budget. We have conducted an ablation study to investigate how varying the size of the training dataset affects model performance. We use the same setting as 3DS-MeanAtten and train models with selected data sizes of 3K, 4K, 5K, 6K, and 7K samples. The results are shown below:
> > > >
> > > > | **Model**          | **Dataset**    | **3k**  | **4k**  | **5k**      | **6k**   | **7k**   |
> > > > |---------------------|----------------|---------|---------|-------------|----------|----------|
> > > > | **baichuan2-7B**    | CMB-Exam       | 29.38   | 30.64   | **31.84**   | 31.50    | 31.54    |
> > > > |                     | MMCU-Medical   | 27.67   | 28.52   | **29.37**   | 28.77    | 29.01    |
> > > > | **baichuan2-13B**   | CMB-Exam       | 46.87   | 47.30   | **47.37**   | 46.95    | 46.98    |
> > > > |                     | MMCU-Medical   | 48.67   | 49.91   | **51.08**   | 50.16    | 50.27    |
> > > > | **Qwen1.5-7B**      | CMB-Exam       | 60.47   | 60.45   | **61.96**   | 60.78    | 60.53    |
> > > > |                     | MMCU-Medical   | 63.64   | 63.92   | **66.09**   | 64.49    | 64.10    |
> > > >
> > > >
> > > > **Performance Peaks at 5K Training Data.** Across all models, when the data budget is increased from 3K to 5K, the model benefits from learning additional domain-aligned data, which improves performance. The best performance is observed with the default 5K data budget. Increasing the data size beyond 5K does not yield further improvements, but leads to slight performance degradation. This may be due to reduced data diversity as more data is added. Redundant, similar data may cause the model to overfit to certain samples and negatively affect fine-tuning performance.
> > > >
> > > > These results align with findings from prior research that show small amounts of high-quality data are sufficient for model alignment (_LIMA_ [16] using 1k samples, _Alpagasus_ [17] using 9k samples) during LLM fine-tuning. Our ablation study empirically shows that 5K is a good budget for domain adaptation fine-tuning.
> > > >
> > > > We have also added this experiment and analysis to **Appendix G.4** of our revised paper.
> > > >
> > > > **(References are attached in Part 5)**

---

> > > > > ### Author Response · Authors · 2024-11-22
> > > > >
> > > > > ## **Q10: Evaluation metrics**
> > > > > Thank you for your kind suggestion. We have added **a detailed explanation of the metrics used in Table 3 (BLEU-1, BLEU-4, and ROUGE) to the Appendix F** of the revised paper. This addition ensures clarification and helps readers better understand our comparisons.
> > > > >
> > > > > ## **Q11: Hyperparameters**
> > > > > Thank you for your kind suggestion. We have added **more details on selection of hyperparameters in Appendix E** of our revised paper. We have clarified the selection process of difficulty thresholds and included other important hyperparameters during training. Our codes and the dataset are also open-sourced to ensure reproducibility.
> > > > >
> > > > > ## **References**
> > > > > [1] Chen, Junying, et al. Huatuogpt-ii, one-stage training for medical adaption of llms. COLM (2024).
> > > > >
> > > > > [2] Petroni, Fabio, et al. Language Models as Knowledge Bases?. EMNLP (2019).
> > > > >
> > > > > [3] AlKhamissi, Badr, et al. A review on language models as knowledge bases. arXiv (2022).
> > > > >
> > > > > [4] Cohen, Roi, et al. Crawling The Internal Knowledge-Base of Language Models. EACL (2023).
> > > > >
> > > > > [5] Yang, Zhaorui, et al. Self-distillation bridges distribution gap in language model fine-tuning. ACL (2024).
> > > > >
> > > > > [6] Yuan, Weizhe, et al. Self-rewarding language models. ICML (2024).
> > > > >
> > > > > [7] Cheng, Daixuan, Shaohan Huang, and Furu Wei. Adapting large language models via reading comprehension. ICLR (2024).
> > > > >
> > > > > [8] Li, Ming, et al. From quantity to quality: Boosting llm performance with self-guided data selection for instruction tuning. NAACL (2024).
> > > > >
> > > > > [9] Gekhman, Zorik, et al. Does Fine-Tuning LLMs on New Knowledge Encourage Hallucinations?. EMNLP (2024).
> > > > >
> > > > > [10] Ren, Mengjie, et al. Learning or self-aligning? rethinking instruction fine-tuning. ACL (2024).
> > > > >
> > > > > [11] Liu, Wei, et al. What makes good data for alignment? a comprehensive study of automatic data selection in instruction tuning." ICLR (2024).
> > > > >
> > > > > [12] Wang, Xidong, et al. Cmb: A comprehensive medical benchmark in chinese. NAACL (2024).
> > > > >
> > > > > [13] Polya, George, and George Pólya. How to solve it: A new aspect of mathematical method. Vol. 34. Princeton university press, 2014.
> > > > >
> > > > > [14] Ramalingam, Dara, Ray Philpot, and Barry McCrae. The PISA 2012 assessment of problem solving. (2017): 75-91.
> > > > >
> > > > > [15] Preheim, Michael, Josef Dorfmeister, and Ethan Snow. Assessing Confidence and Certainty of Students in an Undergraduate Linear Algebra Course. Journal for STEM Education Research 6.1 (2023).
> > > > >
> > > > > [16] Zhou, Chunting, et al. Lima: Less is more for alignment. NIPS (2024).
> > > > >
> > > > > [17] Chen, Lichang, et al. Alpagasus: Training a better alpaca with fewer data. ICLR (2024).

---

> ### Comment · Reviewer_oddX · 2024-11-25
>
> Thank you for the detailed response and for conducting more experiments to substantiate your work. I have increased my score to 6.
>
> A couple of points to note:
>
> * The code repo is accessible now, but the Google Drive link is not.
>
> * I did not see an update regarding the model's inherent knowledge in the updated draft - not sure if I missed it. It might be good to include it in (if not done already) to aid readers.

---

> > ### Author Response · Authors · 2024-11-25
> >
> > ### **Dear Reviewer oddX**,
> > Thank you very much for your recognition of our work and our efforts. We greatly appreciate your thoughtful comments, and address the two points you mentioned:
> >
> > 1. **Google Drive link**: We sincerely apologize for the confusion caused. The issue was due to the closing bracket "]" being inadvertently included in the hyperlink, making it appear inaccessible. The correct link https://drive.google.com/drive/folders/1SfrwQkDrQJ8i_EIqfc2Di0Xa5Y5pzY9H (without the bracket) is functional and accessible.
> > 2. **Model's Inherent knowledge clarification**: Thank you for highlighting this. We have included explanations and supporting references related to the model's inherent knowledge in the Introduction of the newly updated paper (line 057-059). These additions help clarify the concept and provide a better understanding for readers. If there are any specific sections that you feel require further elaboration, please do let us know.
> >
> > We are grateful for your constructive feedback and the opportunity to refine our work. Please don’t hesitate to share additional suggestions if you have any.
> >
> > Best regards,
> >
> > Authors of Paper 10359

---

### Official Review · Reviewer_dZNK · 2024-11-08

**Soundness:** 3
**Presentation:** 3
**Contribution:** 3
**Rating:** 6
**Confidence:** 3

**Summary:**

The paper presents a method to select a subset of the fine-tuning data to fine-tune a language model. Their method is tested on healthcare language data setting where the specificity of the prompts requires models to be fine-tuned in order for the model to perform well. The objective of the data selection algorithm is to find examples which have little noise, are informative, and not too difficult for the model. The authors suggest a two stage algorithm for selection of the samples. The first consists of prompting a language model for useful points and the second looking at the perplexity of prompts, responses and predictions. They fine-tune the model using fine-tuning data that they curated, containing mixtures of different healthcare language data that is composed of 19 million samples. The authors show that their method is the only method that consistently outperforms random acquisition of fine-tuning points.

**Strengths:**

The paper shows that their method clearly outperforms against the benchmarks they tested against. The paper is clearly written, easy to understand, and structured well. The ablation studies related to the size of the model and its ability learn from more difficult points backed up their hypothesis that some points are too difficult for some models to learn, which is one of the motivations for the design of the algorithm.

**Weaknesses:**

## References
Given that selecting fine-tuning points is very relevant to the field of active learning, I was surprised to see zero references to the field unless I missed them, especially since the techniques used in this algorithm are very similar to active learning. Hence the score for presentation, since the work was not contextualised well within prior work. Active learning also deals with quantification of uncertainty of points, much like this paper does using perplexity of models, which I believe is analogous to the uncertainty of a model. There have also been many papers in the field of active learning relating to the topic of finding informative, noise-free and learnable points.
See for example:
Mind Your Outliers! Investigating the Negative Impact of Outliers on Active Learning for Visual Question Answering. Sören Mindermann et al.
Prioritized Training on Points that are Learnable Worth Learning, and Not Yet Learnt. Siddharth Karamcheti et al

There were no references to the metrics BLEU1,4 and Rouge. Even if they are traditional metrics, it would be good to provide a reference for those who are unfamiliar with them. Especially since it is used in one of the two tables that present the main results.

## Further examination of the algorithm
If this paper consisted of just of looking at perplexity of various aspects of the model's perplexity (step 2), then this paper would lack novelty, since this would basically be the same as a standard uncertainty based active learning acquisition method. However, the acquisition algorithm has an additional step (step 1) that directly queries the language model to respond to a set of evaluation criteria designed to rule out low quality examples. This makes the algorithm different from existing active learning methods.
There are some ablation studies in the table showing the removal of different parts of step 2 of the algorithm (D1 to D3). However, there are no ablations for removing completely step 1 or 2 of the algorithm. I would be curious to see what effect of removing completely the step 1 of the algorithm and the effect of removing step 2. I wonder which part of the algorithm is doing the biggest leg work. Couldn't we also prompt the model to ask about its uncertainty of the prompt ? essentially collapsing step 2 of the process in to a few additional prompts in to step 1 of the process.
I would be happy to increase my 'contribution' score and the 'overall' score to 8 if this ablation is added that convinces me that both steps are necessary in the algorithm. i.e. can we get similar results but removing step 1 completely? Can we get similar results by removing step 2 but prompting the model to acquire the uncertainty of prompts and using that as an additional scoring mechanism to score points, removing the need to compute the perplexity of queries. This would add more to the general understanding of what steps are useful when selecting fine-tuning points, which would be a very useful contribution to the field. A confirmation that prompting as well as inspection of perplexity are both integral to the algorithm’s success will be very informative.

## Fairness considerations
Finally, since this paper relates to the healthcare domain where fairness and bias are more important than in other domains, it would be worth mentioning somewhere in the paper the effect that data-selection can have on the fairness of a model and some references to the appropriate literature. Since the paper already has a section on limitations, perhaps the authors could mention that the implications on safety of this method has not been examined and it is left for future work. Some examples of the effects of fine-tuning on models can be found here as a baseline:
The Trade-off between Performance, Efficiency, and Fairness in Adapter Modules for Text Classification. Minh Duc Bui

**Questions:**

How would a practitioner choose the appropriate percentage range of difficulty of prompts for a particular model? How do we know that the model being used is sufficiently large that we can put a higher upper bound for the difficulty scores for the points which will be acquired?

---

> ### Author Response · Authors · 2024-11-22
>
> ## **W1: References**
>
> **Connections Between Active Learning and LLM Data Selection**
>
> Thank you for your detailed feedback and for pointing out the connections between active learning and data selection for LLMs. Your insights are both profound and valuable. Below, we address your comments and clarify how these topics relate to our work.
>
> **We acknowledge that many ideas in active learning are conceptually aligned with techniques in LLM data selection**. For instance, active learning strategies like uncertainty sampling and coreset selection are frequently employed in data selection for LLMs.
>
> * _MATES_ [1] employs data selection by training an influence model to predict the data's validation loss, which shares a similar approach to _Learning Loss for Active Learning_ [2].
> * _MoDS_ [3] calculates changes in loss before and after training on data points; _S2L_ [4] iteratively evaluates a model’s performance on a validation set during training, both reminiscent of loss-based active learning strategies in your cited papers.
>
> While we agree that there are parallels, there are also important distinctions:
>
> 1. **Model Context**: Active learning typically targets small, task-specific models, whereas LLM data selection focuses on general-purpose, pre-trained LLM
>
> 2. **Selection Process**: Active learning dynamically selects data during training (e.g., batch-by-batch), while LLM data selection often occurs entirely before training.
>
> These two areas are generally treated as distinct research directions, and existing works on LLM data selection often do not discuss active learning advancements. However, we appreciate your profound insights into their resemblance. Although due to the limited scope of our study and the page limitations, we can not fully discuss their connections in our current work, we have added references to key active learning works in the revised Related Work section and acknowledge the need for future research.
>
> **Evaluation Metrics (BLEU-1, BLEU-4, ROUGE)**
>
> We appreciate your suggestion. We have included detailed descriptions and references for BLEU-N and ROUGE in **Appendix F** of the revised paper.
>
> **(References are attached in Part 3)**

---

> > ### Author Response · Authors · 2024-11-22
> >
> > ## **W2: Further examination of the algorithm**
> >
> > Thank you for your insightful feedback and suggestions. In response, we have conducted a series of ablation experiments to investigate the contributions of each stage in our proposed two-stage framework and whether Stage 2 (perplexity-based difficulty calculation) could be collapsed into Stage 1.
> >
> > The ablation experiments are conducted as:
> >
> > 1. **Without Stage 1**: Randomly selecting 70,000 samples from the dataset for further difficulty calculation and filtering in Stage 2.
> > 2. **Without Stage 2**: Directly applying k-center sampling to the highest-quality samples identified by Stage 1 without calculating data difficulties.
> > 3. **Collapsing Stage 2 into Stage 1**: Prompting the model to verbally assess the three difficulties (Instruction Understanding, Response Confidence, and Response Correctness) directly, bypassing the original perplexity-based calculation. The prompts are included in Appendix G.1 in the revised paper.
> >
> > The results are presented in the table below:
> >
> > | **LLM Turbo**         | **Baichuan2-7B-Chat** |                  | **Baichuan2-13B-Chat** |                  | **Qwen1.5-7B-Chat** |                  |
> > |-----------------------|------------------------|------------------|--------------------------|------------------|----------------------|------------------|
> > | **Dataset**           | CMB-Exam              | MMCU-Medical     | CMB-Exam                | MMCU-Medical     | CMB-Exam            | MMCU-Medical     |
> > | **Without Stage 1**   | 29.61                 | 28.88           | 44.64                   | 48.06           | 60.37               | 64.03           |
> > | **Without Stage 2**   | 29.41                 | 27.03           | 47.09                   | 50.83           | 61.59               | 65.91           |
> > | **Stage 2 Collapsed into Stage 1** | 29.09                 | 25.97           | 47.28                   | 51.01           | 60.56               | 63.99           |
> > | **_M_ - $MeanAtten$** | **31.84**             | **29.37**       | **47.37**               | **51.08**       | **61.96**           | **66.09**       |
> >
> >
> > From the experiment results, we observe that:
> > * **When Stage 1 is removed, and random samples are provided for Stage 2, performance drops significantly**. This indicates that the initial selection step plays a critical role in identifying high-quality data. Without this step, even the subsequent perplexity-based difficulty calculation cannot fully compensate for the lack of quality filtering.
> > * **The ablation of Stage 2 shows that simply relying on Stage 1 to identify high-quality data is insufficient**. Directly applying k-center sampling on Stage 1's output leads to degraded performance, as it fails to prioritize data with appropriate difficulty levels for fine-tuning.
> > * **Collapsing stage 2 into additional prompts also leads to performance degradation**. During our experiments, we found that models struggle to provide fine-grained assessments of data difficulty when prompted directly. Model outputs were often limited to coarse-grained scores such as 0.5, 0.8 and 1, making it challenging to differentiate data with nuanced levels of difficulty. Additionally, without knowing the model’s exact capabilities, we could not provide in-context learning examples in the prompt to guide finer-grained difficulty judgments. After filtering based on model-prompted difficulty assessments, the resulting dataset typically contained 20,000-30,000 samples from an initial pool of 60,000-70,000. In contrast, **our perplexity-based difficulty calculation provided fine-grained difficulties** and can narrow the selection down to fewer than 10,000 samples. The smaller, more targeted dataset aligns better with the desired moderate difficulty range, as demonstrated by its superior performance in fine-tuning experiments. The performance results demonstrate that **the perplexity-based difficulty calculation in Stage 2 provides essential fine-grained filtering that the model’s verbal assessments cannot currently achieve**. This reinforces the necessity of retaining both steps in the algorithm.
> >
> > We have included these findings and additional insights into **Appendix G.1** in the revised paper, and we hope this addresses your concerns and provides a deeper understanding of the necessity of both stages in our approach.

---

> > > ### Author Response · Authors · 2024-11-22
> > >
> > > ## **W3: Fairness considerations**
> > >
> > > Thank you for your thoughtful comments. We have carefully considered your suggestions and incorporated relevant reflections in **Appendix J**. Below, we address your concerns in detail.
> > >
> > > Our approach employs the LLM to evaluate data quality and calculate data difficulty. **While the evaluation prompt and difficulty calculation method are designed to be neutral**, focusing on medical knowledge and the soundness of the Q&A, we acknowledge that **inherent biases in the underlying base model could possibly influence selection result**. After selection, our method employs adapter modules (LoRA) when fine-tuning the LLM. Therefore, it is susceptible to fairness issues highlighted in the paper you cited: _The Trade-off between Performance, Efficiency, and Fairness in Adapter Modules for Text Classification_, which states that, while adapters generally do not introduce additional bias when base models exhibit minimal bias, their effects can become unpredictable when base biases are more pronounced, potentially amplifying disparities for certain groups. Further research is needed to closely examine the effect of LoRA fine-tuning on LLM's fairness.
> > >
> > > Another limitation stems from the **composition of our training data, which predominantly consists of Chinese medical texts**. While this dataset effectively represents the health conditions of East Asian populations, it may limit the generalizability of the model to other regions or demographics, thereby introducing potential fairness concerns.
> > >
> > > The issue of fairness and bias remains underexplored across the field of data selection for LLM, as most methods—including ours—prioritize model performance on benchmarks for domain knowledge or instruction-following ability, without assessing safety, truthfulness, or fairness on benchmarks like SafetyBench [5] and TruthfulQA [6].
> > >
> > > We have expanded these discussions to **Appendix J** of our revised paper. We also acknowledge the need for future work to investigate the broader impact of data selection for LLM fine-tuning on fairness and safety, especially in sensitive field like healthcare.
> > >
> > > ## **Q1: Selection of difficulty threshos**
> > >
> > > Thank you for your insightful questions regarding the selection of difficulty ranges, and its relation with the model size.
> > >
> > > In our experiments, we evaluated models trained using varying difficulty thresholds based on three different base models, and our findings suggest that the difficulty range 20%-70% is generally a reasonable and effective choice, if not optimal. To better determine the appropriate difficulty thresholds for a specific model, practitioners can use the following procedures:
> > >
> > > 1. **Initial Evaluation**: Before fine-tuning, the base model can be evaluated on a validation set to assess its current domain capabilities. This provides a preliminary understanding of the model's capacity and helps guide the selection of difficulty thresholds. In section 5 Impact of Difficulty Thresholds in our paper, we empirically discover that stronger models typically need more difficult data. Thus, if the model performs well on the domain tasks, a higher difficulty upper bound can be set accordingly.
> > > 2. **Hyperparameter Search**: In our current approach, the difficulty thresholds are treated as hyperparameters. Practitioners can perform a search over potential ranges and select the values that yield the best performance on the validation set. This allows for adapting the difficulty range to the model's specific strengths and weaknesses.
> > >
> > > Finally, we acknowledge that automating the selection of difficulty thresholds is an important direction for future work. Developing methods to dynamically adapt thresholds for different models could enhance the robustness and generalizability of the data selection framework.
> > >
> > > ## **References**
> > >
> > > [1] Yu, Zichun, Spandan Das, and Chenyan Xiong. MATES: Model-Aware Data Selection for Efficient Pretraining with Data Influence Models. NIPS (2024).
> > >
> > > [2] Yoo, Donggeun, and In So Kweon. Learning loss for active learning. CVPR (2019).
> > >
> > > [3] Du, Qianlong, Chengqing Zong, and Jiajun Zhang. Mods: Model-oriented data selection for instruction tuning. arxiv (2023).
> > >
> > > [4] Yang, Yu, et al. SmallToLarge (S2L): Scalable Data Selection for Fine-tuning Large Language Models by Summarizing Training Trajectories of Small Models. NIPS (2024).
> > >
> > > [5] Zhang, Zhexin, et al. Safetybench: Evaluating the safety of large language models. ACL (2024).
> > >
> > > [6] Lin, Stephanie, Jacob Hilton, and Owain Evans. TruthfulQA: Measuring How Models Mimic Human Falsehoods. ACL (2022).

---

> ### Comment · Reviewer_dZNK · 2024-11-25
> **On the additional results**
>
> Thank you for conducting this analysis, I can see that there's variation in how different stages impact different datasets. There is no clear second place for which algorithm works best in this table, but there is a clear first place, being the algorithm proposed. However the difference in these performance numbers are very small, and without error bars its not possible to say that there is a statistical significance, especially since the difference in the best-performing and the second-best-performing algorithm is as low as 1%. As a result, my overall score will remain the same but I will increase my contribution score to 3.

---

> > ### Comment · Reviewer_dZNK · 2024-11-26
> > **On the additional references and fairness considerations**
> >
> > Thank you for adding the additional text regarding the fairness considerations of the paper. This will help to increase awareness of the importance of fairness considerations when fine-tuning large language models.
> > The additional references I believe will embed the paper better in existing fields.
> > However, with respect, I do not consider these adjustments to suffice revising any other scores.

---

> > > ### Author Response · Authors · 2024-11-27
> > > **Regarding the additional results**
> > >
> > > Dear Reviewer dZNK,
> > >
> > > Thank you for your valuable feedback. We greatly appreciate your acknowledgment of our efforts.
> > >
> > > We understand your concerns about the performance gain of our proposed method. While we recognize that the performance differences between the proposed method and the ablation variants are relatively small in the CMB-Exam and MMCU-Medical tasks, our experiments show that **the proposed method shows greater consistency across multiple models**. For instance, although collapsing Stage 2 into Stage 1 results in relatively good performance on _Baichuan2-13B-Chat_, it performs poorly on _Baichuan2-7B-Chat_. In contrast, our method demonstrates **consistent and good performance** across all the models, highlighting its robustness and reliability.
> > >
> > > **To address your concerns and further validate the effectiveness of our designed framework, we have performed additional evaluations on the medical analysis task CMB-Clin**, as we did in the main result part in the paper. We note that in domain adaptation, the aim is to consistently enhance the model's diverse domain abilities. Therefore, it is important for the model to not only provide correct answers in multiple-choice problems but also perform accurate and effective analysis.
> > > The results are shown below:
> > >
> > > | **LLM Turbo**                      | **Baichuan2-7B-Chat**|     |           | **Baichuan2-13B-Chat** |     |           | **Qwen-1.5-7B-Chat**|    |           |
> > > |------------------------------------|---------------------|-------|-----------|----------------------|-------|-----------|-------------------|-------|-----------|
> > > | **Metric**                         | BLEU-1 | BLEU-4    | ROUGE     | BLEU-1    | BLEU-4    | ROUGE     | BLEU-1    | BLEU-4  | ROUGE  |
> > > | **Without Stage 1**                | 17.01  | 38.52     | 19.39     | 14.13     | 29.60     | 16.19     | 15.50     | 31.94   | 15.88  |
> > > | **Without Stage 2**                | 21.29  | 55.74     | 27.62     | 20.56     | 46.86     | 21.83     | 21.55     | 47.39   | 21.55  |
> > > | **Stage 2 Collapsed into Stage 1** | **22.71**  | _60.13_   | _29.46_   | _21.48_   | _50.16_   | _22.69_   | _21.73_   | _52.27_   | _23.41_  |
> > > | **3DS -MeanAtten**                 | _22.61_  | **64.57** | **32.11** | **24.15** | **63.51** | **31.50** | **24.40** | **60.32**   | **28.07**  |
> > >
> > > The results clearly demonstrate that our approach **significantly outperforms** the ablation variants in terms of both BLEU and ROUGE scores. This shows that our method excels in medical analysis, providing more accurate and fluent analysis results.
> > >
> > > We also conduct **GPT-4 judgment** to compare the model's answers to other variants and report the win-rates of our proposed 3DS-MeanAttention. The results are shown below:
> > >
> > > | **LLM Turbo**       | **Baichuan2-7B-Chat**  | |           | **Baichuan2-13B-Chat**  |  |           | **Qwen-1.5-7B-Chat**|  |           |
> > > |---------------------|--------------------|----|-----------|----------------------|--|-----------|---------------|-----|-----------|
> > > | **Metric**          | Win    | Tie    | Lose      | Win    | Tie    | Lose      | Win    | Tie    | Lose      |
> > > | **vs Without Stage 1**  | 65.5   | 12.5   | 22.0      | 66.5   | 9.0    | 24.5      | 70.5   | 3.0    | 26.5      |
> > > | **vs Without Stage 2**  | 65.5   | 11.0   | 23.5      | 66.0   | 15.5   | 28.5      | 66.0   | 5.5    | 28.5      |
> > > | **vs Stage 2 Collapsed into Stage 1** | 62.0   | 9.5    | 28.5      | 63.5   | 18.0   | 18.5      | 54.5   | 2.5    | 43.0      |
> > >
> > > The result demonstrate that the win-rates of our method generally exceeds **60%** across models, further validating its superiority.
> > >
> > > Although the **multiple-choice results** do not provide a definitive ranking of which stage is the most critical, the open-ended question results reveal a clear trend:
> > > - **Removing Stage 1 leads to the worst performance**,
> > > - **Followed by the removal of Stage 2**,
> > > - **Collapsing Stage 2 into Stage 1 results in better performance than both**.
> > >
> > > This suggests that **quality control** (Stage 1) is crucial for ensuring the model provides high-quality responses. Additionally, the importance of **difficulty filtering** (Stage 2) is evident, as even a coarser-grained difficulty measure (Collapsing Stage 2 into Stage 1) still outperforms the method where difficulty is entirely ignored. This pattern emphasizes the importance of **both quality and difficulty control** in optimizing model performance. The above experiments and corresponding explanations are included in our newly revised paper.

---

> > > > ### Author Response · Authors · 2024-11-27
> > > >
> > > > Regarding the **error bars**, we completely understand the importance of statistical significance. However, due to the substantial computational resources and time required for LLMs (data selection, training, and inference) we are unable to conduct the repeated experiments within the time constraints for this revision and rebuttal. This issue is commonly encountered in other works involving LLMs (IDF, Superfiltering, Deita, etc.), where error bars are not typically included due to the high computational cost and time limitations. We apologize for any inconvenience this may cause and hope for your understanding.
> > > > However, we acknowledge the importance of repeated experiments to strengthen the statistical robustness of the results. We will conduct these repeated experiments and include the corresponding statistical information in the camera-ready version if the paper is accepted.
> > > >
> > > > We hope this additional set of experiments and the results provided help to address your concerns regarding the performance.
> > > > We believe that these findings provide stronger evidence for our method's effectiveness and robustness across different models and tasks.
> > > > We sincerely hope that based on the additional experimental results provided, you will kindly reconsider the overall evluation of our paper.
> > > >
> > > > Thank you again for your time and consideration. We look forward to hearing your thoughts on these additional results.
> > > >
> > > > Sincerely,
> > > >
> > > > Authors of Paper 10359

---

> > > > > ### Author Response · Authors · 2024-12-01
> > > > > **Kindly Requesting for Reviewer's Feedback**
> > > > >
> > > > > Dear Reviewer dZNK,
> > > > >
> > > > > We sincerely appreciate your efforts in reviewing our paper and our rebuttal. We understand that you may have a busy schedule, and we are grateful for the time you've dedicated to providing feedback.
> > > > >
> > > > > We have made substantial efforts to address the concerns raised in your follow-up review and we are eagerly awaiting your feedback. As the rebuttal phase is nearing its end, we would be truly grateful if you could share any additional feedback or insights. Should there be any remaining issues or areas requiring further clarification, we are more than happy to address them.
> > > > >
> > > > > Thank you again for your time, consideration and valuable feedback!
> > > > >
> > > > > Sincerely,
> > > > >
> > > > > Authors of Paper 10359

---

### Official Review · Reviewer_4px1 · 2024-11-08

**Soundness:** 3
**Presentation:** 3
**Contribution:** 3
**Rating:** 6
**Confidence:** 2

**Summary:**

The authors introduce a method to address limitations of domain adaptation in LLMs within the medical domain. The method is termed 3DS: Decomposed Difficulty Data Selection. 3DS introduces a better focus on data selection that aligns with the target model’s current knowledge distribution (limiting knowledge gaps), as well as a way to avoid overfitting and allow for increases in difficulty by evolving the data selection to address the changing model.

3DS is divided into two distinct stages 1) Prompt-Driven Data Selection using the target model to select high quality data and 2) Decomposed Difficulty Data Selection to select data of reasonable difficulty to facilitate efficient domain adaption. Difficulty is assessed through three measures: i) Instruction Understanding Difficulty (comprehension assessment using a perplexity score); ii) Response Confidence Difficulty (measured by the model’s conditional perplexity when generating a response given an instruction) and iii) Response Correctness Difficulty (measured by perplexity as well). Additionally, an attention-based importance weighting mechanism is used for ii) and ii) which helps clarify uncertainty in the model further, by adjusting perplexity-based measurements by weighting token according to their semantic importance

3DS outperforms existing methods, which use proprietary LLMs and/or reward models, demonstrating improvements in domain ability (accuracy) by outperforming baselines, as well as claiming improved scalability and generalisability.

**Strengths:**

**Originality:**

- Attempts to build on previous work by swapping out a proprietary LLM with the target model to identify data necessary for alignment

- Motivation for, and the subsequent inclusion of a different metric (attention-based importance weighting mechanism), beyond the commonly used perplexity is supported by evidence from the literature, and in the positive results

**Quality:**
- The approach to the problem of data filtering for (medical) domain adaptation in LLMs is motivated well, and the methods presented in the paper build on previous work and address the limitations indicated in related work (e.g. avoid overfitting, and address increasing data complexity as the model’s knowledge evolves during adaptation)
- The chosen evaluation metrics: perplexity and attention-based and attention-based importance weighting mechanisms, are appropriate for the problem, with the latter addressing some limitations of perplexity which may lead to less important tokens (e.g. prepositions) skewing uncertainty estimates.
- These metrics are used to assess the method's performance against multiple baselines, almost all showing a positive improvement
- Please see the questions for a note on considering an additional metric for assessing excluded out-of-distribution data

**Clarity:**
- The paper generally reads well and is well structured making the background motivations, as well as the methods and results easy to follow.
- Please see comments and suggestions for minor suggestions on improving accessibility

**Significance:**
- Medical domain applications have the potential for significant impact and, despite some concerns about the dataset (see weaknesses), if the dataset is released responsibly, the 1.9 million instances can also have broader applicability and support further research in this domain
- The positive improvement on the baselines is a strong motivator for the beneficial utility of the proposed methods.

**Weaknesses:**

1. Lack of reflection on limitations
- The authors do not engage at all with the potential limitations of their own approach, and how these might be contrasted with “other” methods that use proprietary LLMs. It is my belief that both these approaches are subject to some sort of inherent bias due to bias amplification. Bias amplification would be when one mode that is used in a machine learning pipeline has its own biases being amplified within the downstream task [1].
- More broadly, data filtering is also subject to bias issues, as demonstrated by Hong et al., with CLIP [2]. Without acknowledging these potential limitations, it is very difficult to assess this method’s ability to mitigate these potential, but known effects

2. Dataset curation:
- The authors have not included a Datasheet for Datasets [3] in the appendix. The code and data have not been released: https://anonymous.4open.science/r/3DS-E67F/README.md. The omission of this documentation makes making it even more difficult to assess the following points:

   - The authors claim the data come from open sourced sites, but do not expand on whether the individual terms of services for these sites explicitly allow for this data to be used for machine learning (as one example of a potential terms of service violation)
   - Despite being crawled data, the sensitive nature of the data, particularly the Q&A data crawled from the doctor-patient exchanges, raises ethical concerns [4].

**References:**

[1] Hall *et al.*, 2022. A Systematic Study of Bias Amplification: https://arxiv.org/pdf/2201.11706

[2] Hong *et al.*, 2024. Who's in and who's out? A case study of multimodal CLIP-filtering in DataComp: https://arxiv.org/abs/2405.08209

[3] Gebru *et al.*, 2021. Datasheets for Datasets: https://arxiv.org/abs/1803.09010

[4] Fiesler *et al.*, 2020. No Robots, Spiders, or Scrapers: Legal and Ethical Regulation of Data Collection Methods in Social Media Terms of Service: https://ojs.aaai.org/index.php/ICWSM/article/view/7290/7144

**Questions:**

**Primary questions and suggestions:**
- Was there an ethics review process involved in the development of this dataset? This seems pertinent given the sensitive nature of the data?
- A Datasheet for Datasets [1] can improve transparency around the dataset and highlight considerations for the user, such as how it was curated and expected use cases.
- A deeper consideration of the limitations of this method, particularly with regards to societal impact and potential biases that might arise (e.g. bias amplification through , would strengthen motivations for using this method, as well as inspire future research to address these concerns.
- Have the authors considered the potential limitation of excluding out-of-distribution data that may still be useful downstream? The current metrics treat this out-of-distribution data as ‘noise’, but including an additional evaluation step, such as human-in-the-loop evaluation of the subset data to be excluded could ensure relevant data is still included.

**Minor suggestion for readability:**
- The baselines section is a little difficult to follow without clarification of the baselines used. I believe readability will be improved by introducing the acronyms in more detail there. Equally, the Appendix can be made more accessible by including the citations for the baselines there as well

**References:**

[1] Gebru *et al.*, 2021. Datasheets for Datasets: https://arxiv.org/abs/1803.09010

**Details Of Ethics Concerns:**

The authors have not included a Datasheet for Datasets [1] in the appendix. The code and data have not been released: https://anonymous.4open.science/r/3DS-E67F/README.md. The omission of this documentation and the lack of access to data samples makes it difficult to assess whether this data is safe to release.

There is no information as to whether individual sources that have been scraped have had their respective terms of services checked to see if they allow for the data to be used for machine learning [2]. The data scraped from the consulting sites (patient - doctor Q&A) is of particular concern given the sensitive nature of the topic.

To the best of my knowledge, no ethics review was undertaken by the author's themselves prior to collecting this data.

**References:**

[1] Gebru *et al.*, 2021. Datasheets for Datasets: https://arxiv.org/abs/1803.09010

[2] Fiesler *et al.*, 2020. No Robots, Spiders, or Scrapers: Legal and Ethical Regulation of Data Collection Methods in Social Media Terms of Service: https://ojs.aaai.org/index.php/ICWSM/article/view/7290/7144

---

> ### Author Response · Authors · 2024-11-22
>
> ## **W1: Lack of reflection on limitations**
>
>   We greatly appreciate your insightful feedback. We recognize this as a crucial aspect and address your concerns below. We also incorporated relevant reflections into our revised paper.
>   Our approach employs the LLM to be fine-tuned to assess data quality and difficulty. **While the evaluation prompt and difficulty calcultaion methods are neutral and demongraphic-agnostic, it's true that there is potential bias stemming from the underlying base LLM as you correctly point out and stated in the cited papers**, which may influence data selection. Notably, this limitation is not unique to our approach but also present in existing dara selection methods, where the external LLMs' inherent biases may affect selection results.
>
>   As far as we know, **current data selection methods do not address bias and fairness**. They typically focus on factors like data difficulty (e.g., IFD [1]), quality (e.g., deita [2]) and diversity (e.g., S2L [3]), without examining in detail what data is included or excluded, especially concerning specific demographic groups. Additionally, these methods, including ours, mainly focus on model performance on standard instruction-following, NLU or domain knowledge benchmarks and do not yet incorporate broader tests on LLM safety and truthfulness benchmarks (SafetyBench [4], TruthfulQA [5]), which may also pose certain risks. The impact of data selection and fine-tuning on LLM safety is an underexplored area and needs further research.
>
>   Another potential bias in our method arises from the **training dataset composed of Chinese medical text. This data may better reflect the health conditions of East Asian populations**, which may limit the generalizability to other regions or demographics.
>
>   We agree that your concerns are significant and represent valuable directions for future research. We have added a detailed discussion of these limitations to **Appendix J** in our revised paper.
>
> ## **W2: Dataset Curation**
>
> Thank you for your comments regarding dataset documentation and ethical considerations. We have taken steps to address these concerns and have updated our paper accordingly.
>
> 1. **Code and Model Release:**
> We have now open-sourced the code at https://anonymous.4open.science/r/3DS-E67F and the dataset at https://drive.google.com/drive/folders/1SfrwQkDrQJ8i_EIqfc2Di0Xa5Y5pzY9H ，providing detailed README instructions on usage.
>
> 2. **Datasheet for Datasets**:
> A “Datasheet for Datasets” has been included in the paper‘s Appendix B, providing more detailed descriptions of the dataset construction process, including the sources of the data and sample examples.
>
> 3. **Data Sources and Ethical Considerations**:
> The open-sourced portion of the dataset is built by collecting already publicly available and open datasets that are explicitly permitted for use in training LLMs. For the crawled data, it was obtained from a publicly accessible website belonging to our university's partner hospital. We understand the sensitivity of Q&A data derived from doctor-patient dialogues and anonymize all personal or identifiable information thoroughly. Additionally, permission for its use in this research was obtained from the institution.
> We believe these updates and clarifications enhance the transparency and ethical accountability of our dataset. Thank you again for raising these important points, which have helped us improve our documentation and discussion.
>
> ## **Q1: Regarding Ethics Review**
>
> As noted in our response to Weakness-2, the datasets collected are open-sourced and publicly available for medical LLM training. Specifically, the crawled data was obtained from a partner hospital’s public website with explicit approval, and all data was anonymized to ensure privacy. These steps adhered to ethical guidelines and institutional approvals.
>
> ## **Q2: A Datasheet for Datasets**
>
> As stated above, we have included a datasheet for our dataset in **Appendix B** of our revised paper. This datasheet details the motivation, the collection and curation processes, as well as provides data examples for transparency and clarity.
>
> ## **Q3:  Limitations of the Method**
>
> We have addressed these concerns in detail in our response to Weakness-1 and incorporated the relevant discussion into the **Appendix J** of our revised paper. We acknowledge the importance of societal impact and potential biases and will consider these concerns in future work.
>
> **(References are attached in Part 2)**

---

> > ### Author Response · Authors · 2024-11-22
> >
> > ## **Q4: Potential Limitation of Excluding OOD data and Human-in-the-loop**
> >
> > We agree that some excluded data might still be relevant to downstream tasks and could potentially improve model performance. However, the goal of our method is to design **an autonomous data filtering process** that minimizes human intervention and reduces the associated costs. By relying on the model itself to select data, we acknowledge that there is a **trade-off** between automation and ensuring that no valuable data is discarded.
> >
> > We appreciate your suggestion that incorporating a human-in-the-loop evaluation step could help retain useful out-of-distribution data. This is indeed a valuable direction for future research and exploration.
> >
> > ## **Q5: Baseline Readability**
> >
> >   We apologize for the lack of readability in the baselines section and thank you for your helpful feedback. In response, we have revised the section, emphasizing the baseline acronyms in bold formatting to improve readability and include detailed explanations of the acronyms. Additionally, we have updated the baseline section in **Appendix D** to include citations for better accessibility.
> >
> > ## **References**
> >
> > [1] Li, Ming, et al. From quantity to quality: Boosting llm performance with self-guided data selection for instruction tuning. NAACL (2024).
> >
> > [2] Li, Ming, et al. Superfiltering: Weak-to-strong data filtering for fast instruction-tuning. ACL (2024).
> >
> > [3] Liu, Wei, et al. What makes good data for alignment? a comprehensive study of automatic data selection in instruction tuning. ICLR (2024).
> >
> > [4] Yang, Yu, et al. SmallToLarge (S2L): Scalable Data Selection for Fine-tuning Large Language Models by Summarizing Training Trajectories of Small Models. NIPS (2024).
> >
> > [5] Zhang, Zhexin, et al. Safetybench: Evaluating the safety of large language models. ACL (2024).
> >
> > [6] Lin, Stephanie, Jacob Hilton, and Owain Evans. TruthfulQA: Measuring How Models Mimic Human Falsehoods. ACL (2022).

---

> > > ### Author Response · Authors · 2024-11-26
> > > **Kindly Requesting for Review of Our Rebuttal**
> > >
> > > Dear Reviewer 4px1,
> > >
> > > As the last day that authors can make revision to the paper is approaching, we extend our gratitude once more for your valuable and insightful comments!
> > >
> > > We have provided careful and detailed responses to all your questions and it would be very helpful if you could confirm whether we have fully answered them. If you have any further questions or suggestions, we would be grateful to receive them so that we can address them before the revision deadline, should any additional revisions be necessary.
> > >
> > > Best regards,
> > >
> > > Authors of Paper 10359

---

> ### Comment · Reviewer_4px1 · 2024-11-26
> **Thank you for addressing the comments, but I still have concerns around the limitations and the Datasheet**
>
> Thank you very much for your response and updates to the paper. I appreciate the time the authors put into responding to my points, and the effort behind making updates to the paper. I believe my concerns have been addressed to a degree, and so Ihave increased my score to 6.
>
> I believe that the technical implementation is well motivated, however, as the authors state in their own rebuttal - no other work is addressing fairness. I think given the breadth and depth already given to optimising the factors such as difficulty, quality etc. of data selection, moving the needle on fairness as well improving performance is extremely important and should be given more attention in all works. I strongly believe this is a direction that should be taken more seriously, given the significant potential for impact.
>
> While the additional sections do outline the limitations, I’d also encourage that these have a more central part in the paper, rather than being relegated to the appendix. I think it’s reasonable to assume most people don’t engage as deeply with the appendix as the main paper and centring these concerns can also help inspire other work to be more considerate for fairness applications.
>
> **Follow up questions about the DataSheet:**
> While I appreciate the steps taken to include this, I'd still like to ask:
> 1. Why did you choose this particular version of a Datasheet? Would you mind sharing your reference, please?
> There is a standardised version [1] which I believe is an important and rigorous documentation template. By filling in all these details, it actually helps other researchers use the data. For example, the example usage section can help lower the barrier to entry for other researchers to use this dataset as intended, and specificity around motivations and data composition provides a deeper insight into the data itself contributing to more rigorous research practices.
> 2.I’m also curious why there are no efforts for maintenance plans? I would imagine at the very least the authors should be open to fixing errata, and removing content that is found to be problematic?
>
> Releasing a dataset to influence model development is a big responsibility so I do believe the authors should acknowledge this in the effort that it takes to do so - in terms of clear and thorough documentation and a commitment to maintenance plans. It is not my intention to discourage the releasing of the dataset, but rather encourage that it is done responsibly so that the broader field can in fact benefit.
>
> **Minor suggestions:**
> Please proofread the newly added sections. Some typos have snuck in.
>
> **Reference**
> [1] Gebru, T., Morgenstern, J., Vecchione, B., Vaughan, J. W., Wallach, H., Iii, H. D., & Crawford, K. (2021). Datasheets for datasets. Communications of the ACM, 64(12), 86-92.

---

> > ### Author Response · Authors · 2024-11-27
> >
> > Dear Reviewer 4px1:
> >
> > We sincerely thank you for your acknowledgement for our efforts and your detailed feedback and constructive comments.
> > Below is a point-by-point response to your comments and follow up questions:
> >
> > #### 1. **Fairness and Bias in Data Selection**
> > We acknowledge that this is an extremely critical and timely research direction that requires more attention, particularly as the community generally does not take considerations. Due to time and space constraints, we were unable to include a more in-depth discussion of fairness and bias in the current version. However, we want to emphasize that we consider these issues to be of paramount importance, and plan to adjust the paper to address more on these concerns within the main body of the paper in the camera-ready version if it is accepted, emphasizing their importance to inspire more thoughtful and responsible research in this field. Additionally, we strongly agree that fairness in data selection for large LLMs is an important topic, and we hope to explore this direction in future work.
> >
> > #### 2. **Datasheet for Datasets**
> > We largely followed the guidelines provided in your referenced paper. As our dataset is not collected from scratch but instead comprises a curated selection of existing open-source datasets, we focused on addressing several key aspects from the referenced paper that we felt were most pertinent to our dataset’s context.
> >
> > However, we acknowledge the importance of comprehensive and standardized documentation. We have added a "Usage Case" section in the newly revised paper, and we will continue to refine the Datasheet, revising and expanding it in the camera-ready version to ensure that it follows the standardized template more thoroughly.
> >
> > #### 3. **Maintenance Plans for the Dataset**
> > We appreciate your concern about dataset maintenance and agree that it is an important consideration. Initially, we assumed that the dataset, being a broad, comprehensive instruction fine-tuning resource, would allow developers to select the data that best fits their models. However, we now realize that this perspective was overly simplistic and have revised our stance accordingly.
> >
> > We are now committed to maintaining and updating the dataset, including correcting any errors and addressing possible problematic content if reported. We will also provide more detailed documentation for users, ensuring that we can offer support and make corrections as necessary. We have updated the maintenance part in the datasheet accordingly.
> >
> > #### 4. **Minor Suggestions**
> > Thank you for pointing out the typos. We have proofread the revised text and made corrections. We will continue to revise the content and the writing in the camera-ready version upon acceptance.
> >
> > Once again, we greatly appreciate your feedback and valuable suggestions.
> >
> > Sincerely,
> >
> > Authors of Paper 10359

---

> > > ### Comment · Reviewer_4px1 · 2024-11-28
> > > **Updated soundness score to 3**
> > >
> > > Dear authors
> > >
> > > Thank you for your positive and productive engagement with the feedback. Please note I have increased my soundness score from 2-->3, however at this time I will keep my overall score at 6.
> > >
> > > I appreciate the openness to adapting the message, and putting in effort to complete the full Datasheet. I absolutely understand page and time constraints, but hopefully we can all iteratively focus on these and help build the right kind of momentum.

---

> > > > ### Author Response · Authors · 2024-12-01
> > > > **Thanks for acknowledging our response**
> > > >
> > > > Dear Reviewer 4px1,
> > > >
> > > > Thank you so much for your time to review our work and responses. We deeply appreciate your thoughtful feedback and guidance that are very valuable to our research.
> > > >
> > > > Sincerely,
> > > >
> > > > Authors of Paper 10359

---

### Note · Authors · 2025-01-23

I have read and agree with the venue's withdrawal policy on behalf of myself and my co-authors.